# Convex Markov Games: A New Frontier for Multi-Agent Reinforcement Learning

Ian Gemp [1]  Andreas Haupt [2]  Luke Marris [1]  Siqi Liu [1]  Georgios Piliouras [1]

## Abstract

Behavioral diversity, expert imitation, fairness, safety goals and others give rise to preferences in sequential decision making domains that do not decompose additively across time. We introduce the class of convex Markov games that allow general convex preferences over occupancy measures. Despite infinite time horizon and strictly higher generality than Markov games, pure strategy Nash equilibria exist. Furthermore, equilibria can be approximated empirically by performing gradient descent on an upper bound of exploitability. Our experiments reveal novel solutions to classic repeated normal-form games, find fair solutions in a repeated asymmetric coordination game, and prioritize safe long-term behavior in a robot warehouse environment. In the prisoner's dilemma, our algorithm leverages transient imitation to find a policy profile that deviates from observed human play only slightly, yet achieves higher per-player utility while also being three orders of magnitude less exploitable.

## 1. Introduction

Modern solutions to (perfect-information) multi-agent reinforcement learning (MARL) typically build upon the Markov game (MG) model (Littman, 1994), a generalization of the classical single agent Markov decision process (MDP) to the multi-agent setting (see Table 1, Linear Loss).

Similarly, the MDP has provided the foundation for development of single agent RL research. However, modern solutions to single agent RL now build upon the convex MDP (cMDP), an extension of the MDP that allows for more expressive preferences over agent behavior that do not decompose additively across time (Zhang et al., 2020).

[1]Google DeepMind, London, UK [2]College of Computing, MIT, Cambridge, MA, USA. Correspondence to: Ian Gemp <imgemp@google.com>.

*Proceedings of the 42$^{nd}$ International Conference on Machine Learning*, Vancouver, Canada. PMLR 267, 2025. Copyright 2025 by the author(s).

| # of Agents | Linear Loss | Convex Loss (Concave $u$) |
|:---:|:---:|:---:|
| $n = 1$ | MDP [1] | convex MDP [2] |
| $n > 1$ | MG [3] | (**Ours**) convex MG |

*Table 1.* We introduce the convex Markov Game (cMG) to fill a gap at the intersection of multi-agent and reinforcement learning research. We demonstrate how to use cMGs to model a variety of utilities $u$ for a linear reward function $r$: (a) Creativity, using an entropy bonus, (b) Imitation, using a divergence from a reference occupancy measure, (c) Fairness, using a state-wise penalty, and (d) Safety, with a non-smooth loss. This paper establishes existence of equilibria for such utilities, proposes a (sub)differentiable loss for its computation, and demonstrates its use in selecting desirable multi-agent behaviors in each setting, at times by tracing a homotopy from a cMG *back* to an MG. *References*: [1] Bellman (1966); [2] Zahavy et al. (2021); [3] Shapley (1953).

Mathematically, convex MDPs allow agent goals to be expressed as general convex functions of their long run occupancy measure (the frequency with which the agent takes a given action in a given state). An exemplar is the goal of maximizing the entropy of an agent's occupancy measure to encourage exploration (Hazan et al., 2019); imitation, risk-aversion, robustness, and other goals are also common (Mutti et al., 2022; García & Fernández, 2015). In the case where the goal is expressed as a linear function of the occupancy measure, cMDPs reduce to MDPs.

Naturally, MARL researchers have adopted this generalization empirically in their work, for example to encourage agent curiosity and the discovery of new and interesting strategies in games like Chess (Zahavy et al., 2023). We remark (†) on the advantages of these more general objectives in selecting desireable equilibria in experiments.

However, there has yet to be a formal definition of a framework extending cMDPs to the multi-agent setting (Table 1). More troubling is the the lack of any analysis proving Nash equilibria, the central solution concept in multi-agent and game theory research, even exist in this setting.

In this work, we formally define the convex Markov game (cMG) and prove the existence of pure Nash equilibria by appealing to topological arguments. From a theoret-

ical perspective, this result is interesting as properties of cMGs break the assumptions leveraged by prior work on Markov games (e.g., convex best response correspondences and existence of value functions). In addition, we derive a (sub)differentiable upper bound on the exploitability of a policy profile that can be efficiently computed and optimized. Lastly, we describe several use cases for cMGs beyond just curiosity, including imitation, fairness, and safety. Using our derived upper bound, we solve for approximate Nash equilibria in each of these settings, showing improved performance over sensible baselines. In some cases, we construct a deformation (homotopy) from a cMG to an MG that allows us to derive police profiles that despite being similar to observed human play, exhibit higher welfare and orders of magnitude lower exploitability.

The article is structured as follows. We define the class of convex Markov games in section 2. We show that pure-strategy Nash equilibria exist in section 3. Our analysis of a loss for equilibrium computation is in section 4, which we put to practice in our experiments in section 5. We discuss related literature in section 6 and conclude in section 7.

## 2. Convex Markov Games

The main definition of this article is the *convex Markov Game* (cMG).

**Definition 1.** *A convex Markov game is given by a 6-tuple* $\mathcal{G} = \langle \mathcal{S}, \mathcal{A} = \times_{i=1}^{n} \mathcal{A}_i, P, u, \gamma, \mu_0 \rangle$:

- *Players* $i = 1, 2, \ldots, n$,
- *a finite state space $\mathcal{S}$ and initial distribution $\mu_0 \in \Delta^{\mathcal{S}}$,*
- *finite action spaces $\mathcal{A}_1, \mathcal{A}_2, \ldots, \mathcal{A}_n$[1],*
- *a state transition function $P \colon \mathcal{S} \times \mathcal{A} \to \Delta^{\mathcal{S}}$,*
- *a discount factor $\gamma \in [0, 1)$, and*
- *a set of real-valued utilities, $u_i(\mu_i, \pi_{-i})$, each continuous and concave in player $i$'s occupancy measure, $\mu_i \in \Delta^{\mathcal{S} \times \mathcal{A}_i}$, and continuous in each policy $\pi_{j \neq i}$.*

**The Policy View** Players $i = 1, \ldots, n$ choose (*stationary*) policies $\pi_i \colon \mathcal{S} \to \Delta^{\mathcal{A}_i}$. A policy profile $\pi = (\pi_1, \pi_2, \ldots, \pi_n)$ induces a sequence of states and joint actions $(s_t)_{t \in \mathbb{N}}$ and $(a_t)_{t \in \mathbb{N}}$, and a state-action occupancy measure

$$\mu^\pi(s, a) = (1 - \gamma) \sum_{t=0}^{\infty} \gamma^t \mathbb{P}(s_t = s, a_t = a | \mu_0, \pi, P).$$

We can recover state-action occupancies from a matrix equation. Let $P^\pi$ be the state transition matrix under $\pi$. Then the state occupancy measure $\mu^s(s) = \sum_a \mu(s, a)$ can be written as a function of $\pi$ in matrix notation as

$$\mu^s(\pi) = (1 - \gamma) \Big( [I - \gamma P^\pi]^{-1} \mu_0 \Big), \qquad (1)$$

---

[1] Action spaces can be generalized to have state-dependence.

and the state-action occupancy can be recovered as $\mu(s, a) = \mu^s(s) \cdot \pi(a|s)$. The marginal on this probability for only agent $i$'s actions can be recovered as

$$\mu_i(\pi_i, \pi_{-i}) = (1 - \gamma)([I - \gamma P^\pi]^{-1} \mu_0 \mathbf{1}_{|\mathcal{A}_i|}^\top) \odot \pi_i, \quad (2)$$

where $\odot$ denotes the Hadamard product and $\pi_{-i}$ indicates all policies except player $i$'s. Hence, we can also write $u_i(\mu_i(\pi_i, \pi_{-i}), \pi_{-i})$ purely with policies as $u_i(\pi_i, \pi_{-i})$.

Players may choose to randomize over stochastic policies. We say that a player's strategy $\rho_i$ is a *mixed-strategy* if it is a distribution over policies. For example, let $\pi_i^{(a)}$ and $\pi_i^{(b)}$ both be stochastic policies. Then an example of a mixed-strategy is one in which player $i$ plays $\pi_i^{(a)}$ and $\pi_i^{(b)}$ with equal probability $1/2$. If $\rho_i$ puts all mass on a single policy $\pi_i$, we call $\rho_i$ a *pure-strategy* and write $\pi_i$ directly.

We say that a (random) policy profile $\rho = (\rho_1, \rho_2, \ldots, \rho_n)$ is a *(mixed-strategy) Nash equilibrium* if for all players $i$,

$$\mathbb{E}_{\pi \sim \rho}[u_i(\pi_i, \pi_{-i})] \geq \mathbb{E}_{\pi_i' \sim \rho_i', \pi_{-i} \sim \rho_{-i}}[u_i(\pi_i', \pi_{-i})],$$

for any policy $\rho_i'$. If $\rho$ is a pure strategy profile as defined above, we call it a *pure-strategy Nash equilibrium*.

We call any policy $\tilde{\pi}_i$ (or occupancy measure $\tilde{\mu}_i$) player $i$'s *best response* to a profile $\rho$ if it achieves an expected utility that is maximal among all policies (respectively, occupancy measures). Note best responses are the solutions to each individual player's MDP with all other player policies held fixed, and so they necessarily *can* exist as pure strategies: $\tilde{\pi} \in \arg\max_{\pi_i'} \mathbb{E}_{\pi_{-i} \sim \rho_{-i}}[u_i(\pi_i', \pi_{-i})]$

**The Occupancy Measure View** In our analysis and the algorithms proposed in this article, it will be helpful to frame and solve problems directly in the space of occupancy measures $\mathcal{U}$.

Given opponent policy profiles $\pi_{-i}$ we can define the probability kernel $P_i^{\pi_{-i}}$ as

$$P_i^{\pi_{-i}}(s'|s, a_i) = \sum_{a_{-i} \in \mathcal{A}_{-i}} P(s'|s, a_i, a_{-i}) \prod_{j \neq i} \pi_j(a_j|s).$$

Given these, we can reformulate an individual agent's decision problem as

$$\max_{\mu_i \in \mathbb{R}^{|\mathcal{S}| \times |\mathcal{A}_i|}} u_i(\mu_i, \pi_{-i}) \qquad (3)$$

$$s.t. \quad \mu_i \geq 0 \qquad (4)$$

$$\sum_{a_i \in \mathcal{A}_i} \big( I - \gamma P_i^{\pi_{-i}}(\cdot|\cdot, a_i) \big) \mu_i(\cdot, a_i) = (1 - \gamma)\mu_0 \quad (5)$$

where $P_i^{\pi_{-i}}(\cdot|\cdot, a_i)$ is a next-state by state transition matrix and $\mu_i(\cdot, a_i)$ is a vector denoting the probability of taking action $a_i$ in every state. Because $u_i$ are concave in $\mu_i$,

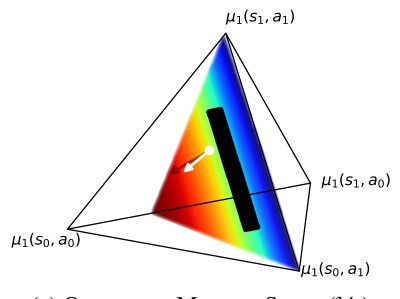

(a) Occupancy Measure Space ($\mathcal{U}_1$)

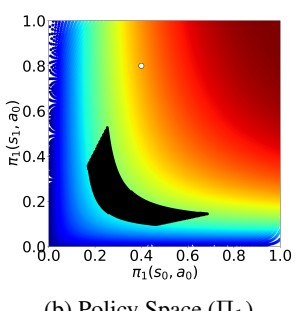

(b) Policy Space ($\Pi_1$)

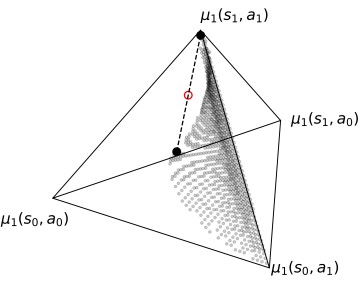

(c) Joint Occupancy Measure Space ($\mathcal{U}$)

*Figure 1.* (A convex Markov Game) A 2-player, 2-state, 2-action convex Markov game. Colors help visualize the effect of the nonlinear transformation from occupancy space to policy space (7). Full environment details are in Appendix F.2. (a) The feasible set of occupancy measures for player 1 given a fixed policy for player 2 with best response region shown in black. Player 1's occupancy gradient $\nabla^i_{\mu_i}$ points off the 3-simplex so is not shown. Instead, we show $\nabla^i_{\mu_i}$ projected onto the tangent space of the feasible set in white, i.e., $\Pi_{T\mathcal{U}_i}(\nabla^i_{\mu_i})$. The vector shadowing the white one is $\nabla^i_{\mu_i}$ projected onto the tangent space of the 3-simplex; note this vector points off the feasible set (into the page). (b) A 2-d slice of player 1's feasible policy space. The set of player 1's best response policies to $\pi_2$ (white dot) in black is non-convex when viewed in policy space. (c) We consider the joint occupancy measure space $\mathcal{U}$ where $\pi_1 = \pi_2$ (implies $\mu_1 = \mu_2$). Black dots indicate $\mu_1$ where the maximum Bellman flow constraint violation (5) is less than 0.01. The line connects two points from the feasible set whose midpoint (in red) lies outside the set, revealing the non-convexity of $\mathcal{U}$.

and given the linearity of the constraints in (4) and (5), this problem is convex, motivating the name of *convex* Markov Games (cMGs). If $u_i(\mu_i, \pi_{-i}) = r_i(\pi_{-i})^\top \mu_i$ where $r_i$ is a flattened state-action reward vector and $\mu_i$ is similarly flattened, we recover Littman's Markov game framework (as mentioned in Table 1), however cMGs allow a much wider class of utilities (e.g., ones that include entropy of the occupancy measure). Notice that the feasible set in (5) is defined by constraints that depend on $\pi_{-i}$. For convenience, we define the so-called *action correspondence*, $\mathcal{U}_i = \mathcal{M}_i(\pi_{-i})$, which maps every profile of opponent policies to the feasible set of occupancy measures for player $i$, i.e, problem (5) can be succinctly written

$$\max_{\mu_i \in \mathcal{M}_i(\pi_{-i})} u_i(\mu_i, \pi_{-i}). \tag{6}$$

We will use the feature that policies can be *partially* recovered from occupancy measures as

$$\pi_i(\mu_i)(a|s) = \begin{cases} \frac{\mu_i(s,a)}{\sum_{a'} \mu_i(s,a')} & \text{if } \sum_{a'} \mu_i(s,a') > 0 \\ \text{any } \pi_i^s \in \Delta^{|\mathcal{A}_i|} & \text{otherwise.} \end{cases} \tag{7}$$

## 3. Existence of Nash Equilibrium

We first show that under fairly general assumptions, mixed Nash equilibria exist. We then argue that for the purposes of learning, pure-strategy Nash equilibria are particularly desirable, and show that these exist as well.

**Proposition 1.** *Mixed-strategy Nash equilibria exist in convex Markov Games.*

This statement follows easily by using the policy view.

Each player's optimization problem in terms of policies is:

$$\max_{\pi_i \in (\times_{s=1}^{|\mathcal{S}|} \Delta^{\mathcal{A}_i})} u_i(\mu_i(\pi_i, \pi_{-i}), \pi_{-i}). \tag{8}$$

Appendix B shows that all $u_i$ are continuous, differentiable functions of $(\pi_i, \pi_{-i})$, and since the strategy sets are compact metric spaces, the existence of mixed-strategy NE follows from classical existence results (Glicksberg, 1952; Fudenberg & Tirole, 1991, Theorem 1.3, p. 35).

While existence of mixed-strategy equilibria are a descriptively helpful tool, they have limitations for applications in learning. In particular, learning a continuous distribution over stochastic policies would practically require function approximation or discretizing the space, greatly increasing the complexity of the learning problem; this challenge has been broached but not sufficiently solved in the context of, for example, generative adversarial networks (Arora et al., 2017). Therefore, we provide another statement that ensures the existence of pure-strategy Nash equilibria.

**Theorem 1.** *Pure-strategy Nash equilibria exist in convex Markov Games.*

The proof of this statement is in Appendix A and relies on topological arguments, in particular the *contractibility* of best response sets (Debreu, 1952; Kosowsky, 2023). While best response sets are convex in occupancy measure space, they are non-convex in policy space (Figure 1a-b). This breaks the assumptions of the Kakutani (1941) fixed point theorem, which provides the foundation for many equilibrium existence results in multi-agent games including cMG's parent framework of Markov games (Fink, 1964), and is one reason why we must appeal to more general theory here.

A set is *contractible* if there exists a continuous homotopy (deformation) that shrinks that set to a point. Intuitively, if best response sets are contractible then they behave topologically similarly to points, and so one would expect the principles of the Brouwer (1911) fixed point theorem that Nash (1950) famously applied in his own existence proof should apply. In fact, all convex sets are contractible, and so Kakutani's fixed point theorem also follows from this argument. Despite their non-convexity, best response sets in cMGs are contractible, which the reader can confirm visually for the example in Figure 1b.

One additional obstruction to the analysis of cMGs is the forfeiture, in the underlying cMDPs, of the recurrence relation known as the Bellman equation, which gives rise to the notion of a value function. In fact, Fink (1964) used the existence of value functions along with Kakutani's fixed point theorem to prove existence of NE in Markov (stochastic) games. We circumvented this obstruction with more general equilibrium arguments, but the lack of access to traditional value functions presents a further obstacle to designing efficient algorithms which we broach next.

## 4. Computation of Equilibria

While equilibria exist, even computing NEs in (vanilla) MGs is PPAD-hard (Rubinstein, 2015; Daskalakis et al., 2023)—since cMGs generalize MGs, computing NEs in cMGs is at least as hard. Nevertheless, we present a practical gradient-based approach for approximating equilibria that we find works well empirically.

The most common notion of approximation for NEs is *exploitability* ($\epsilon$), defined as

$$\epsilon = \max_{i=1,\dots,n} \epsilon_i \tag{9}$$

$$\text{where } \epsilon_i = \max_{z \in \mathcal{M}_i(\pi_{-i})} u_i(z, \pi_{-i}) - u_i(\mu_i, \pi_{-i}). \tag{10}$$

Exploitability measures the most any player can gain by unilaterally deviating to another policy (equiv., occupancy measure). Mechanistically, in cMGs, it corresponds to each player solving problem (6)—a constrained, convex optimization problem—and then reporting the value they attained beyond that of their strategy under the approximate equilibrium profile $\pi$. In particular, a policy profile $\pi$ with vanishing exploitability ($\epsilon = 0$) is a pure-strategy Nash equilibrium.

**High Level Approach**: It should also be clear from the max in (10) that exploitability (9) is always non-negative. Therefore, one can imagine solving for an NE (where $\epsilon = 0$) by minimizing exploitability, i.e., using it as a loss function. Exploitability is relatively expensive to compute as it requires solving $n$ convex programs, so we instead derive a cheap upper bound whose gap is controlled by

a hyperparameter $\tau$. Exploitability (as well as our upper bound) is non-convex and hence naive gradient descent is not guaranteed to find a global minimum. However, in what follows, we leverage "temperature annealing" ideas that have been successful in several other game classes (normal-form/extensive-form/Markov) and find them beneficial empirically for cMGs as well.

The next result extends that of Gemp et al. (2024) from the normal-form game setting to show that exploitability is bounded from above by a constant depending on the action space size, and a projected gradient (compare the white vector to the one behind it in Figure 1a). The following result bounds the exploitability of a cMG with utilities $u_i$ using players' utility-gradients of a cMG with a small amount of entropy regularization, i.e. $u_i^\tau(\mu_i, \pi_{-i}) = u_i(\mu_i, \pi_{-i}) + \tau H(\mu_i)$, where $H$ denotes Shannon entropy.

**Theorem 2** (Low Temperature Approximate Equilibria are Approximate Nash Equilibria). *Let $\nabla_{\mu_i}^{i\tau}$ be player $i$'s entropy regularized gradient and $\pi$ be an approximate equilibrium of the entropy-regularized game with $\tau > 0$. Then,*

$$\epsilon_i(\pi) \leq \tau \log(|\mathcal{S}||\mathcal{A}_i|) + \sqrt{2}\|\Pi_{T\mathcal{U}_i}(\nabla_{\mu_i}^{i\tau})\|,$$

*where $\Pi_{T\mathcal{U}_i}$ is a projection onto the tangent space of $\mathcal{U}_i$, and $\nabla_{\mu_i}^{i\tau}$ is the gradient of $u_i^\tau$ with respect to $\mu_i$.*

We can give more intuition for the analysis of the projection operator. If $A(\pi_{-i}) \in \mathbb{R}^{|\mathcal{S}| \times (|\mathcal{S}| \cdot |\mathcal{A}_i|)}$ such that $A(\pi_{-i})\mu_i = (1 - \gamma)\mu_0$ represents the linear equality constraints in (5), then the projection matrix is given by

$$\Pi_{T\mathcal{U}_i(\pi_{-i})} = I_{|\mathcal{S}| \times |\mathcal{S}|} - A^\top (AA^\top)^{-1} A. \tag{11}$$

For ease of notation, we omit the dependence of $A$ on $\pi_{-i}$. $\Pi_{T\mathcal{U}_i(\pi_{-i})}$ is differentiable if and only if $A(\pi_{-i})$ has full row rank (i.e., rank $|\mathcal{S}|$). Lemma 1 in Appendix B shows that $A(\pi_{-i})$ has full row rank.

Inspired by the bound in Theorem 2, we define the following projected-gradient loss function for cMGs:

$$\mathcal{L}^\tau(\pi) = \sum_i \|\Pi_{T\mathcal{U}_i}(\nabla_{\mu_i}^{i\tau})\|^2. \tag{12}$$

As a consequence of Theorem 2, we obtain Corollary 1

$$\epsilon(\pi) \leq \tau \log(|\mathcal{S}| \max_i |\mathcal{A}_i|) + \sqrt{2n\mathcal{L}^\tau(\pi)}. \tag{13}$$

We can directly minimize $\mathcal{L}^\tau$ over the space of policy profiles $\pi$. Each player's policy is subject to simplex constraints that are independent of the other players. In this way, we combine the two distinct strengths of the policy and occupancy measure views. In the policy view, player strategy sets ($\Pi_i$) are independent and convex, making for simple independent updates and projections back to the feasible sets. In the occupancy measure view ($\mathcal{U}_i$), we enjoy

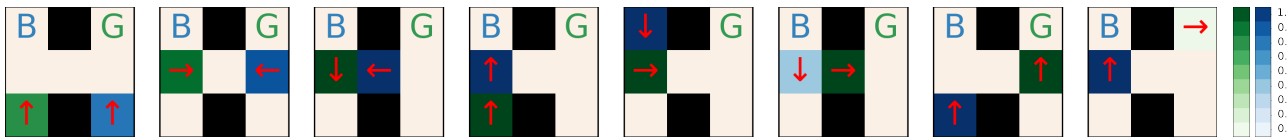

*Figure 2.* (Approximate `PGL` Equilibrium) Arrows indicate each player's most likely action (4 cardinal directions & "Stay") with their color indicating their probability. Moves to the same location are awarded randomly to one of the agents. The goal state is marked by B and G where green achieves the top right corner and blue, the top left, simultaneously. Note that in last grid, $\pi(\downarrow) = 0$.

convexity of player losses which enable derivation of upper bounds on exploitability.

Note that our proposed loss is composed of a projection operator $\Pi_{T\mathcal{U}_i}$, the gradients of our concave utilities $u_i$, and the mapping from policies to occupancy measures (2). All of these components are differentiable assuming that the utilities are also differentiable[2].

This yields a differentiable loss and allows for optimization using automatic differentiation. If we represent agent policies in an unconstrained space with *logits*, we can then minimize $\mathcal{L}^\tau$ directly with respect to agent policies using our preferred unconstrained optimizer "Opt", e.g., Adam. In addition, we repeatedly anneal $\tau$, intended to mimic standard protocols for normal-form, extensive-form, and (vanilla) Markov games (McKelvey & Palfrey, 1995; 1998; Gemp et al., 2022; Eibelshäuser & Poensgen, 2023). We refer to this approach as *projected-gradient loss* minimization (`PGL`), with pseudocode in Algorithm 1.

---

**Algorithm 1** Projected-Gradient Loss Minimization (`PGL`)

---

1: Given: Initial profile $\pi$, temperature schedule $\tau_t$
2: **for** $t = 0, \ldots, T$ **do**
3:    $\pi \leftarrow \text{Opt}(\pi, \nabla_\pi \mathcal{L}^{\tau_t})$
4: **end for**
5: Output: $\pi$

---

Note that Algorithm 1 does not come with any convergence guarantees. Although we take inspiration from previous approaches which carefully trace a continuum of equilibria throughout the annealing process (Turocy, 2005), cMGs introduce a distinct challenge to precisely replicating this family of homotopy methods. The family of homotopy (annealing) methods we imitate rely on first solving for the equilibrium of a transformed game. In prior game classes, this meant solving for the maximum entropy profile which is easy (all players play uniform strategies). In cMGs, even this step is complex. While the objective of maximizing the sum of the entropy of each player's occupancy measure is concave, the feasible set of joint occupancy measures is non-convex (see Figure 1c). Hence, even defining the starting point for a homotopy process that imitates prior approaches is difficult.

---

[2]Strict differentiability of utilities is not required by *autodiff* libraries, e.g., JaX (Bradbury et al., 2018).

## 5. Experiments

We test a variety of *non*linear utilities in several domains. In the first set of (creativity-based) domains, we compare against four baseline algorithms, and discuss and contrast the resulting exploitability and policy profiles. The resulting experiments demonstrate other distinct use cases of cMGs.

**Baselines.** The first baseline, $\min \epsilon$, directly minimizes exploitability using a differentiable convex optimization package CVXPYLAYERS in JAX (Agrawal et al., 2019; Bradbury et al., 2018). In the second baseline, Sim, all players simultaneously run gradient descent on their losses with respect to their policies and we report the performance of the running average of the policy trajectory. Policies at each state are represented in $\mathbb{R}^{|\mathcal{A}_i|-1}$ as a softmax over $|\mathcal{A}_i| - 1$ logits with the last logit fixed as $0$. In the third, RR, agents alternate gradient descent steps in round-robin fashion. We also compare against the SGAME-SOLVER (Eibelshäuser & Poensgen, 2023) package of homotopy methods for Markov games.

**Hyperparameters.** We minimize $\mathcal{L}^{\tau_t}(\pi)$ with Adam; its internal state is not reset after annealing. Three types of annealing schedules $\tau_t$ are used for entropy regularization (Appendix E). Each policy $\pi_i$ is initialized to uniform unless otherwise specified. All experiments except *pathfinding* were run on a single CPU and take about a minute to solve although exact exploitability reporting via CVX-OPT (Diamond & Boyd, 2016) increases runtime approximately $10\times$; pathfinding used 1 GPU.

**Domains.** We consider seven domains: one synthetic, two grid worlds, and four iterated normal-form games (NFGs). The first is a multi-agent pathfinding problem. The second domain is the classic two-player, iterated prisoner's dilemma (compare Tucker & Straffin Jr (1983)) where agents may choose to cooperate or defect with their partner. The third domain is a three-player, public goods game where agents may choose to contribute all or none of their savings to a public pool which is then redistributed evenly with a growth multiplier of 1.3 (compare Janssen & Ahn (2003)); payoff is measured in terms of player profits. In the fourth, we consider a three-player El Farol bar

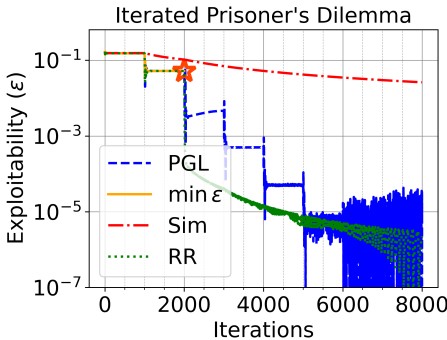 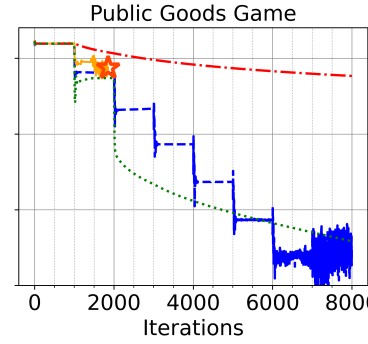 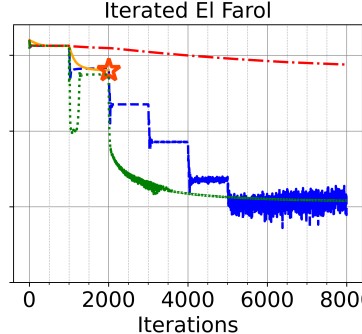

*Figure 3.* (Convergence to Nash equilibria) `RR` and `PGL` descent yield the lowest exploitability yet converge to different equilibria. `PGL` drops coincide with temperature annealing. $\min \epsilon$ crashes, marked by stars. All algorithms performed best with a learning rate of 0.1

problem where players choose whether to go to a bar or stay home (Arthur, 1994). Agents receive maximum payoff (2) for attending an uncrowded bar (fewer than 3 people), followed by staying home (1), followed by attending a crowded bar (0). The fifth domain is the classic Bach-Stravinsky game where agents must coordinate to attend a performance despite different preferences. The synthetic domain was presented in Figure 1 and is described in Appendix F.2. Lastly, we consider a robotic warehouse inspired grid world. We set $\gamma = 0.99$ in all domains. Iterated NFGs use the last joint action selected by each player as state, i.e., $\mathcal{S} = \mathcal{A}$.

### 5.1. Creativity

Our first application considers utilities that value solutions that cover more than a small subset of the state-action space, which leads to "creative" equilibria, as we show. We get such outcomes for utilities that incorporate entropy bonuses for high (Shannon) entropy occupancy measures, $u_i(\mu_i, \pi_{-i}) = r_i(\pi_{-i})^\top \mu_i + \tau H(\mu_i)$ where $r_i(\pi_{-i}) = \mathbb{E}_{a_{-i} \sim \pi_{-i}}[r_i(s, a_i, a_{-i})]$ as an abuse of notation.

To find equilibria of the original game, we anneal the weight on this entropy bonus towards zero, emulating prior work on *homotopy* methods for equilibria in Markov games (Eibelshäuser & Poensgen, 2019).

First, we consider a multi-agent pathfinding problem. Two agents must coordinate to pass through a bottleneck doorway on their way to a joint goal state. The reward for reaching the goal state is 100 for both players; $-0.01$ reward otherwise. Upon reaching the goal state, the agents are reset to the start state (leftmost grid in Figure 2). Our algorithm returned an approximate equilibrium where the final utility for each agent was 24.5 and the exploitability was 1.7 (approximately 7% of their utility).

A single rollout of the final learned policy is shown in Figure 2. The agents race to cross the doorway, after which,

one agent takes the center position and the other steps aside. In the third and fourth to last frames, the blue agent moves downward due to the small remaining entropy bonus. Both agents then move upward towards the goal state. In the final frame, green executes a "no-op" action to the right as blue moves into goal position. Despite learning a factorized Nash equilibrium policy profile, the agents exhibit coordinated actions at certain steps (e.g., Figure 2, frame 3). This coordination is achieved through observations of partner players' grid locations, but richer coordination is theoretically possible with richer observation spaces.

Next, we examine three iterated NFGs. Figure 3 shows our algorithm has vanishing exploitability for all of them. In each game, directly minimizing exploitability in CVX-PYLAYERS crashes due to numerical instabilities. Round-robin descent exhibits similar qualitative behavior to our method, and faster convergence than simultaneous descent.

For a study of creativity, it is valuable to inspect the equilibria the methods found. Round-robin descent converges to an asymmetric NE in iterated El Farol where one player goes to the bar every night, while the two other players alternate. Like `PGL`, SGAMESOLVER converges to the symmetric, state-independent NE of the underlying NFG (attending the bar with probability 0.707). In IPD and IPGG, round-robin and SGAMESOLVER similarly converge to the state-independent NE policies of the underlying NFGs (DD for IPD, zero contribution for IPGG). In contrast, our approach reveals more nuanced, symmetric policies in IPD and IPGG which we explain and discuss below.

**Remark** (†): The entropy of a player's occupancy measure is different from the entropy of their policy; the latter only measures entropy of action distributions in each state, ignoring the distribution across states. Interestingly, this difference manifests in the structure of the equilibria we discover. At high entropy, agents must explore the entire state space, which includes joint cooperation in the iterated prisoner's dilemma (IPD) and joint donation in the iterated

| State $s = (a_1^{t-1}, a_2^{t-1})$ | $a^*$ | $\pi_1(a^*\|s)$ |
|:---:|:---:|:---:|
| (C, C) | C | 0.73 |
| (C, D) | D | 0.74 |
| (D, C) | C | 0.66 |
| (D, D) | D | 0.81 |

*Table 2.* (Approximate `PGL` Equilibrium on IPD) The best response is denoted $a^* = \arg\max_a \pi_1(a\|s)$; $C$ denotes Cooperate; $D$ denotes Defect. The utility per-player is 0.47 compared to that of other classic IPD strategies: tit-for-tat (0.5), win-stay, lose-shift (0.67), grim trigger (0.42), defect-defect (0.33).

| State $s = (a_1^{t-1}, a_2^{t-1}, a_3^{t-1})$ | $a^*$ | $\pi_1(a^*\|s)$ |
|:---:|:---:|:---:|
| (None, None, None) | None | 0.99 |
| (None, None, All-In) | None | 0.66 |
| (None, All-In, None) | None | 0.66 |
| (None, All-In, All-In) | All-In | 0.60 |
| (All-In, None, None) | None | 0.80 |
| (All-In, None, All-In) | All-In | 0.56 |
| (All-In, All-In, None) | All-In | 0.56 |
| (All-In, All-In, All-In) | All-In | 0.86 |

*Table 3.* (Approximate `PGL` Equilibrium on IPGG) The best response is denoted $a^* = \arg\max_a \pi_1(a\|s)$. Per-player utility is 0.03 versus 0 for zero contribution policies.

| State $s = (a_1^{t-1}, a_2^{t-1})$ | $a^*$ | $\pi_1(a^*\|s)$ | $\pi_1^h(a^*\|s)$ |
|:---:|:---:|:---:|:---:|
| (C, C) | C | 0.83 | 0.86 |
| (C, D) | D | 0.52 | 0.65 |
| (D, C) | D | 0.53 | 0.55 |
| (D, D) | D | 0.86 | 0.87 |

*Table 4.* Approximate Nash equilibrium recovered by our algorithm in IPD after annealing KL regularization to the human policy reported in the rightmost column (Romero & Rosokha, 2023, Table 1, Current, Direct-Response). The best response is denoted $a^* = \arg\max_a \pi_1(a\|s)$; $C$ denotes Cooperate; $D$ denotes Defect. The utility per-player under our learned symmetric policy profile is 0.48 versus 0.46 for the human policy. In addition, our learned policy profile is $1.4 \times 10^{-4}$-exploitable at every state, whereas the human policy profile is substantially more exploitable, being 0.47-exploitable over any initial state. Exploitability over all initial states may be seen as an analogue of Markov perfection in Markov games (Maskin & Tirole, 2001).

public goods game (IPGG). As temperature is annealed, the players receive less of a bonus for exploration, however, this transient introduction to mutually beneficial play has a lasting impact on equilibrium selection.

In IPD, `PGL` finds a symmetric policy, shown in Table 2, that is quite similar to the famous *tit-for-tat* policy. If both players cooperated on the last round, then they are likely to continue cooperating. If player 2 defected, then player 1 is likely to defect (even more likely if 1 defected previously). If 1 defected and 2 cooperated, 1 is actually more likely to cooperate in the next round, in an act of reciprocation.

In IPGG, `PGL` finds a symmetric policy as well, shown in Table 3. In contrast to the zero contribution policies found by all other methods, `PGL` finds a policy that, intuitively, is more likely to contribute funds when other agents do.

### 5.2. Imitation

Our second application considers utilities that value policies similar to policies observed in human experiments. In this experiment, we build on the homotopy experiment in the creativity section where we annealed our entropy coefficient $\tau$, but instead of annealing entropy, we anneal a KL penalty to the human state-action occupancy measure, $u_i(\mu_i, \pi_{-i}) = r_i(\pi_{-i})^\top \mu_i - \tau d_{\mathrm{KL}}(\mu_i \| \mu_i^{\mathrm{ref}})$.

**Remark** (†): Matching occupancy measures is akin to matching long-run trajectories whereas matching policies does not necessarily match trajectories when other agents adjust their policies during learning as is the case here.

The reference human policies $\mu_i^{\mathrm{ref}}$ were derived from experiments where subjects played the iterated prisoner's dilemma, selecting cooperate (C) or defect (D) in each period (Romero & Rosokha, 2023, Table 1, Current, Direct-Response). Subjects were required to confirm their opponent's action after each period. This ensured that they were capable of representing a policy that conditions on the previous action. Table 4 reports the symmetric policy we learned while regularizing to the human occupancy mea-

sure. Note that this new policy profile has slightly higher utility for all agents and is very close to an equilibrium, independent of the starting state.

After two scenarios where we used a sequence of cMGs to discover creative and human-like equilibria, we consider a setting in which the utilities are not annealed, but are constant across time.

### 5.3. Fairness

We now consider utilities that value fair visitation of states. In the Bach-Stravinsky game, two players must choose whether to attend a performance by Bach or Stravinsky. If they misalign, they get zero reward. However, one player prefers Bach to Stravinsky (3 *vs.* 2), whereas the other player prefers Stravinsky to Bach (3 *vs.* 2). We incorporate a term into both player's objectives that penalizes any difference in long-run attendance of Bach versus Stravin-

sky to incentivize fair, equal attendance of the two shows, $u_i(\mu_i, \pi_{-i}) = r_i(\pi_{-i})^\top \mu_i - (\sum_{a \in \{S,B\}} \mu_i((B,B),a) - \sum_{a \in \{S,B\}} \mu_i((S,S),a))^2$.

We initialize logits for both players' policies with a standard normal. We set temperature $\tau$ to zero and then optimize with a learning rate of 0.1 for 1000 iterations.

In 10 random trials, both players converge to the same approximate NE where they vote for their favored event 60% of the time regardless of their actions on the previous day. The maximum exploitability $\epsilon$ over the 10 trials is $2.5 \times 10^{-5}$, and the max difference between $\sum_a \mu_i((B,B),a)$ and $\sum_a \mu_i((S,S),a)$ is $2.14 \times 10^{-5}$, implying this is a "fair" behavioral profile by our fairness metric.

### 5.4. Safety

Lastly, we explore applications where a "safe" long-run behavior of the multi-agent system is desired (Miryoosefi et al., 2019). Our algorithm is able to find an exact NE in the synthetic convex loss domain (i.e., $\epsilon = 0$) discussed in the motivating example of Figure 1. Recall the loss for each agent is zero if their $\mu_i$ lies in the given "safe" region marked in black. There are no conventional "rewards" $r_i$ in this synthetic domain. See Appendix F.2 for details.

In addition, we demonstrate a safety application with a grid world where two robots pick up and drop off packages in a warehouse. Packages can only be picked up at a central location where it is potentially dangerous for the robots to move quickly if they happen to share the pickup space simultaneously. At the same time, they are incentivized to drop off as many packages as possible.

Agents maximize their discounted return minus a convex safety loss, which penalizes them for the frequency they take the *fast* action in the pickup state beyond 10%: $100 \cdot \max\left(0, \mu_i(s = (\text{pickup}, \text{pickup}), a = \text{fast}) - 0.10\right)$. In other words, any frequency below 10% is deemed sufficiently safe, but beyond that a linear penalty is applied.

Our algorithm is able to find an approximate NE in this warehouse domain with and without the safety loss, achieving low exploitability, $\epsilon \leq 3.4 \times 10^{-2}$ and $1.0 \times 10^{-3}$ respectively. In either domain, the learned policy always chooses the fast action in all states except for when both agents are picking up a package. When the safety loss is **not** included, the agents select to move fast 69% of the time. **With** the safety loss, fast is chosen 42% of the time reflecting the convex penalty for unsafe behavior.

## 6. Related Work

Our work relates to the single-agent literature on convex Markov games and equilibrium selection in Markov games. We rely on NE existence-proof techniques from topology, loss minimization for equilibrium computation, and use homotopy methods as inspiration for our experiments. Finally, we unify approaches to creativity, imitation, fairness, and safety from multi-agent learning.

**Convex Markov Decision Processes** Markov decision processes (MDPs) are the predominant framework for modeling sequential decision making problems, especially in infinite-horizon settings (Puterman, 2014, §6.9). The goal of a decision maker in an MDP is typically to maximize a $\gamma$-discounted sum of rewards earned throughout the sequential decision process. In the infinite-horizon setting, recent research has exploited an alternative, but equivalent view of maximizing the expected reward under the agent's stationary state-action occupancy measure (Zhang et al., 2020)—the probability of being in a given state and taking a given action. This viewpoint reveals an optimization problem with a linear objective (maximize return) and linear constraints (valid occupancy measure); from this launchpad, research has generalized to convex objectives that incorporate, for example, the (neg)entropy of the occupancy measure in order to maximize exploration of the MDP (Zahavy et al., 2021) or maximize robustness (Grand-Clément & Petrik, 2022). Research on cMDPs has recently surged, however, formulations and solutions to *nonstandard Markovian control problems* have a long history (Kallenberg, 1994; Takács, 1966).

A number of works have explored solutions to MDPs with similarly complex objectives and in a variety of settings. For example, prior research has looked beyond concave utilities to multi-objective (Cheung, 2019b) or submodular objectives (Prajapat et al.; De Santi et al., 2024). Mutti et al. (2022) pointed out practical concerns with the infinite-trials assumption *baked-in* to the standard convex MDP formulation, motivating the study of finite trials (Mutti et al., 2023). Others have considered the non-stationary (Marin Moreno et al., 2024b) and online (Cheung, 2019a) settings as well. Designing scalable algorithms for convex MDPs is a challenge. Zhang et al. (2020) proposed a model-free (variational) policy gradient approach for general utilities that was then combined with variance reduction techniques for improved performance in subsequent work (Barakat et al., 2023). Marin Moreno et al. (2024a) developed an efficient model-based RL approach for the finite horizon setting, and Geist et al. (2022) shed light on a connection to mean-field games, enabling the design of new algorithms for cMDPs. This mean-field games connection also inspired other work to extend inverse-RL to cMDPs where the aim is to uncover an agent's utility function from observed behavior (Çelikok et al., 2024). Recent work developed a practical, supervised learning approach for the continuous time setting applicable to training flow and diffusion models (De Santi et al.).

**Markov Games.** When multiple agents' decision making problems interact, the MDP framework is extended to a Markov game, also known as a stochastic game (Thuijsman, 1997; Littman, 1994). In the game setting, we seek equilibria, a notion of simultaneous optimality for all agents. Fink (1964) proved the existence of stationary (time-independent) $\gamma$-discounted Nash equilibria in $n$-player, general-sum stochastic games. A homotopy approach that traces the continuum of quantal response equilibria performs well at approximating Nash equilibria in the limit of zero temperature (Eibelshäuser & Poensgen, 2019). Other approaches are tailored for more restricted two-player, zero-sum settings (Daskalakis et al., 2020; Goktas et al., 2023) or settings where agent incentives are a priori aligned such as Markov potential games (Leonardos et al., 2021). Prior work proved a negative result for value iteration based approaches stating it is not possible to derive a stationary equilibrium policy from Q-values in general Markov games (Zinkevich et al., 2005); however, in IPD specifically, self-play Q-learners converge to *win-stay, lose-shift* (Bertrand et al., 2025). Lastly, recent work extends the Markov game, a Markov chain of normal-form games, to a chain of *abstract economies* (or *pseudo-games*) (Goktas et al., 2025), games with jointly constrained strategy spaces. In contrast, our cMG can almost be seen as generalizing a Markov game to an abstract economy, however, this leads to an unnatural interpretation of unilateral deviations in terms of occupancy measures.

**Techniques.** Our proposed loss function extends that designed in recent work for normal-form games (NFGs) (Gemp et al., 2024) to the convex Markov game setting. In their work, the focus was on constructing a loss amenable to unbiased estimation. In NFGs, the feasible set $\mathcal{U}_i = \Delta^{\mathcal{A}_i}$ is fixed, independent of other players' strategies. This allows the construction of an unbiased estimator of their loss assuming access to unbiased gradients of player's utilities. Our loss applies to a more general class of games, but sacrifices unbiasedness.

Our applications anneal the temperature of an entropy bonus and/or a Kullback-Leibler divergence penalty. Such approaches relate to homotopy continuation-based approaches to equilibrium computation (Harsanyi et al., 1988). McKelvey & Palfrey (1995) introduced quantal response equilibria along with a homotopy from infinite to zero temperature defining their limiting logit equilibrium in normal-form games (and also extensive-form (McKelvey & Palfrey, 1998)). More recent work extended this approach to Markov games (Eibelshäuser & Poensgen, 2019).

**Applications.** Our goals of creativity, imitation, fairness, and safety are not new to multi-agent applications. Zahavy et al. (2022; 2023) leveraged convex MDPs to discover more creative play in Chess. Bakhtin et al. (2022); Jacob et al. (2022) used KL-regularization towards human play to recover strategically superior policies in Diplomacy. Hughes et al. (2018) models inequity aversion in complex MARL domains. And Shalev-Shwartz et al. (2016) forgoes the Markovian assumption altogether to tackle safe autonomous driving. Zamboni et al. (2025) formulates a cMG with identical payoffs to specifically target group exploration in, for example, robotics domains. In contrast to these applications, our framework of convex Markov games allows for a unified analysis of several domains using a common language and algorithmic principles.

## 7. Conclusion

Convex Markov Games are a versatile framework for multi-agent reinforcement learning. The cMG framework induces equilibria that exhibit diverse state visitation, emulate human data, optimize notions of fairness, or avoid unsafe system states. Not only do pure-strategy equilibria exist despite non-convex best response correspondences, there is also a differentiable upper bound that can be minimized to find them. In several domains, we show how deforming a cMG over training can help to pick out novel approximate equilibria with properties more desirable than those found by baseline techniques such as higher welfare, symmetry, reciprocation, and state-dependent behaviors.

Our work opens up several directions for future research including theoretical questions around other possible solution concepts (e.g., correlated or coarse-correlated equilibria) as well as studying more specific game classes under more restrictive assumptions (e.g., potential or identical payoff settings). In a similar vein, we have, in concurrent work, constructed an efficient, convergent, model-free approach to solving cMGs in the two-player, zero-sum setting (Kalogiannis et al., 2025).

Under the $n$-player, general-sum setting we investigate in this work, we proposed a centralized approach that assumes knowledge of the transition dynamics. If dynamics are estimated, we pointed out issues obtaining unbiased estimates of the projection operator $\Pi_{T\mathcal{M}_i(\mu_{-i})}$. A model-free, decentralized training approach could scale to more complex domains and richer policies.

The entropy of agents' occupancy measures is already utilized in multi-agent applications, particularly in robotics domains to enhance multi-agent *exploration* (Burgard et al., 2000; Rogers et al., 2013; Tan et al., 2022; Zamboni et al., 2025). Unrelatedly, the tuning of large language models (LLMs) to align with human feedback has been recently formulated as a Markov game (Wu et al., 2025). We hope cMGs can help provide a framework from which to better understand these and other problems in the future.

## Acknowledgements

We are grateful to Conrad Kosowsky for helpful discussions on topological equilibrium existence topics.

## Impact Statement

This paper presents work whose goal is to advance the field of multi-agent reinforcement learning. There are many potential societal consequences of our work, none which we feel must be specifically highlighted here.

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

# A. Equilibrium Proof

$$u_i(\mu_i, \pi_{-i}) = \langle r_i, \mu_i \rangle - 100 \cdot \max\left(0, \mu_i(s = (\text{pickup}, \text{pickup}), a = \text{fast}) - 0.10\right) \tag{14}$$

**Theorem 3** (Corollary 8, Kosowsky (2023))**.** *Suppose that $(\Pi_i)$ is a finite collection of compact and connected manifolds, and suppose we have continuous utility functions $u_i : \Pi \to \mathbb{R}$ with $\Pi = \times_{i=1}^{n} \Pi_i$ satisfying the following properties for every player $i$:*

1. *Player $i$'s best response is a continuous function $\text{BR}_i : \Pi_{-i} \to C(\Pi_i)$ where $C$ is the set of nonempty, closed subsets of $\Pi_i$ (equipped with the upper Vietoris topology), i.e., $\text{BR}_i$ is a compact-valued, upper-hemicontinuous correspondence;*

2. *Every output value $\text{BR}_i(\pi_{-i})$ is contractible and has a contractible neighborhood in $\Pi_i$;*

3. *There exists a homotopy that takes $\text{BR}_i$ to a constant map and whose output values —which are elements of $C(\Pi_i)$ —are all contractible and have a contractible neighborhood in $\Pi_i$.*

*Then the game has a pure-strategy Nash equilibrium.*

**Theorem 1.** *Pure-strategy Nash equilibria exist in convex Markov Games.*

*Proof.* This proof uses the Nash equilibrium existence result in Corollary 8 by Kosowsky (2023) (restated as Theorem 3 above). For a slightly simpler proof that follows Debreu (1952), see Theorem 5 below.

We will first confirm the premise of Theorem 3 holds for convex Markov games. In cMGs, each player's strategy set $\Pi_i$ is the simplex-product, $\times_{s \in \mathcal{S}} \Delta^{|\mathcal{A}_i|-1}$, a compact, convex set. All convex sets are connected. The joint strategy set is simply the product space of the player's individual strategy sets matching Theorem 3: $\Pi = \times_{i=1}^{n} \Pi_i$. In addition, definition 1 of cMGs assumes each $u_i$ is continuous in players' occupancy measures $\mu_j$ for all $j$. Lemma 2 proves each $\mu_j$ is a differentiable (hence continuous) function of $\pi = (\pi_1, \ldots, \pi_n)$, therefore continuity of $u_i$ in occupancy measures implies continuity in policies.

We will next prove each property required by Theorem 3 in order.

Player $i$'s feasible set in policy space is the simplex-product $\Pi_i$ (regardless of $\pi_{-i}$). Player $i$'s occupancy measure $\mu_i(\pi_i, \pi_{-i})$ is continuous in $\pi_{-i}$, hence the feasible set $M_i(\pi_{-i})$ in occupancy space (i.e., the image of $\mu_i(\pi_i, \pi_{-i})$ under $\Pi_i$ for each $\pi_{-i}$) is continuous in $\pi_{-i}$. The set $M_i(\pi_{-i})$ is also always non-empty (any policy is always feasible and we can map any policy to an occupancy measure) and compact (it is the intersection of a hyperplane with the simplex). Recall player $i$'s utility function $u_i$ is continuous in $\mu_i$. By Berge's maximum theorem (Aliprantis & Border, 2006, Theorem 17.31), the best-response occupancy set for player $i$ is upper hemicontinuous in $\pi_{-i}$ with non-empty and compact values. The mapping from occupancies to policies is upper hemicontinuous. The composition of upper hemicontinuous maps remains upper hemicontinuous (Aliprantis & Border, 2006, Theorem 17.23). Therefore, $\text{BR}_i$ is a compact-valued, upper-hemicontinuous correspondence satisfying property 1.

Moving to the next property, every player's best (occupancy measure) response problem is a convex optimization problem, whose solutions always form a convex set. Convex sets are contractible (this is where concave utilities play the most critical role in the proof). The mapping from occupancies to policies is upper hemicontinuous. The set of best (policy) responses therefore remains contractible after mapping. Lastly, all of $\Pi_i$ can serve as a contractible neighborhood for each $\text{BR}_i$, confirming property 2.

To satisfy the last property, define $H(t, \pi_{-i}) = \text{BR}_i((1-t)\pi_{-i} + t\pi^0_{-i})$, satisfying the endpoint conditions where $\pi^0$ is any valid strategy profile, e.g., uniform. $\Pi_{-i}$ is convex so every linear interpolation between $\pi_{-i}$ and $\pi^0_{-i}$ is in $\Pi_{-i}$. Recall, $\text{BR}_i$ is upper hemicontinuous. The interpolated strategy is obviously upper hemicontinuous in both $t$ and $\pi_{-i}$. Therefore, their composition forming $H$ proves $H$ is upper hemicontinuous in $t$ and $\pi_{-i}$. Recall that every best response set is contractible with a contractible neighborhood. Every set returned by $H(t, \pi_{-i})$ is a $\text{BR}_i(\pi'_{-i})$ for some $\pi'_{-i} \in \Pi_{-i}$. Hence $H(t, \pi_{-i})$ is contractible with a contractible neighborhood for every $t$ and $\pi_{-i}$ completing the claim. $\qquad \square$

**Theorem 4** (Debreu (1952))**.** *Suppose that $(\Pi_i)$ is a finite collection of contractible polyhedra, and suppose player $i$'s choice of action $\pi_i \in \Pi_i$ is further restricted to a non-empty, compact subset $\Pi'_i(\pi_{-i}) \subseteq \Pi_i$. Also denote $\Pi = \times_{i=1}^{n} \Pi_i$*

*and suppose we have utility functions $u_i : \Pi \to \mathbb{R}$ continuous on $\Pi'_i(\pi_{-i})$ for all $\pi_{-i}$ satisfying the following properties for every player $i$:*

1. *Player $i$'s utility at their best response, $\max_{\pi_i \in \Pi'_i(\pi_{-i})} u_i(\pi_i, \pi_{-i})$ is a continuous function;*

2. *Every output value $\texttt{BR}_i(\pi_{-i})$ is contractible.*

*Then the game has a pure-strategy Nash equilibrium.*

**Theorem 5.** *Pure-strategy Nash equilibria exist in convex Markov Games.*

*Proof.* Each $\Pi_i$ is a simplex product, hence a contractible polyhedron. In cMGs, $\Pi'_i(\pi_{-i}) = \Pi_i$ for all $\pi_{-i}$, so Theorem 4 is actually more general than what is needed here. Each $u_i$ is continous in $\pi_i$ by Lemma 2. By Berge's maximum theorem (Aliprantis & Border, 2006, Theorem 17.31), player $i$'s utility at their best response is a continuous function. Lastly, each $\texttt{BR}_i$ is contractible as already proven for property 2 in Theorem 1. $\qquad\square$

## B. Occupancy from Policy is Differentiable

**Lemma 1.** *The Bellman flow constraint matrix has full row-rank ($|\mathcal{S}|$) and is fixed independent of other player policies.*

*Proof.* Note the Bellman flow constraints can be written in matrix form as

$$\sum_{a \in \mathcal{A}_i} (I_{s' \times s} - \gamma P_{i,a}) \mu_{i,a} = (1 - \gamma) \mu_0 \tag{15}$$

where $P_i = P(s'|s, a)$ denotes the $\mathcal{S} \times \mathcal{S} \times \mathcal{A}_i$ tensor of transition probabilities and $P_{i,a} = P_{i,a}(s'|s)$ selects out a single action, leaving an $\mathcal{S} \times \mathcal{S}$ matrix.

This constraint can be written without the $\sum_{a \in \mathcal{A}_i}$ by constructing the rectangular block matrices

$$I_{s' \times (sa)} = \begin{bmatrix} I_{s' \times s}, \ldots, I_{s' \times s} \end{bmatrix} \tag{16}$$

and

$$P_{i, s' \times (sa)} = \begin{bmatrix} P_{i, a_1}, \ldots, P_{i, a_m} \end{bmatrix}. \tag{17}$$

Then

$$(I_{s' \times (sa)} - \gamma P_{i, s' \times (sa)}) \mu_i = (1 - \gamma) \mu_0 \tag{18}$$

We can examine the first $s' \times s$ block of $(I_{s' \times (sa)} - \gamma P_{i, s' \times (sa)})$ and show that this matrix is full rank, i.e., of rank $|\mathcal{S}|$. If this matrix is full rank, then its rows are linearly independent. Extending our view to the full matrix, i.e., all columns, cannot render any of these original rows linearly dependent.

Note that the first block is represented by $(I_{s' \times s} - \gamma P_{i,a})$ for some action $a$. Using the Gershgorin circle theorem, we can bound the eigenvalues of this matrix to lie in a union of circles which all exclude the origin. Consider any column $c$, then every circle has a center in $\mathbb{R}_+$. In addition, the leftmost point of every circle lies in $\mathbb{R}_+$:

$$(1 - \gamma P_{i,a}(c|c) - \sum_{s' \neq c} |\gamma P_{i,a}(s'|c)| \tag{19}$$

$$= (1 - \gamma P_{i,a}(c|c) - \gamma \sum_{s' \neq c} P_{i,a}(s'|c) \tag{20}$$

$$= 1 - \gamma \sum_{s'} P_{i,a}(s'|c) \tag{21}$$

$$= 1 - \gamma > 0. \tag{22}$$

Therefore, this matrix is non-singular, i.e., full-rank. There are only $|\mathcal{S}|$ rows, hence the row-rank cannot increase, which proves the claim. $\qquad\square$

**Lemma 2.** *Player $i$'s state-action occupancy measure $\mu_i$ is a differentiable (and hence continuous) function of the player policies $\pi$.*

*Proof.* Recall

$$\mu_i(\pi) = (1-\gamma)\Big([I - \gamma P^\pi]^{-1}\mu_0 \mathbf{1}_{a_i}^\top\Big) \odot \pi_i \tag{23}$$

where

$$P^\pi(s', s) = \langle P_j^{\pi_{-j}}(s', s, :), \pi_j(s, :) \rangle \tag{24}$$

and

$$P_j^{\pi_{-j}}(s', s, :) = \sum_{a_{-j}} P(s'|s, a_i, a_{-j}) \prod_{k \neq j} \pi_k(s, a_k) \tag{25}$$

so

$$P^\pi(s', s) = \sum_{\boldsymbol{a}} P(s'|s, \boldsymbol{a}) \prod_j \pi_j(s, a_j). \tag{26}$$

Then,

$$
\begin{aligned}
\frac{\partial \mu_i(x, y)}{\partial \pi_j(x', y')} &= (1-\gamma)\Big[[I - \gamma P^\pi]^{-1}\mu_0(s)\Big]_x \mathbb{1}(x = x', y = y', j = i) \\
&\quad + (1-\gamma)\frac{\partial}{\partial \pi_j(x', y')}\Big([I - \gamma P^\pi]^{-1}\Big)(\mu_0 \mathbf{1}_{a_i}^\top) \odot \pi_i
\end{aligned}
\tag{27}
$$

where

$$\frac{\partial}{\partial \pi_j(x', y')}\Big([I - \gamma P^\pi]^{-1}\Big) = \gamma[I - \gamma P^\pi]^{-1}\frac{\partial P^\pi}{\partial \pi_j(x', y')}[I - \gamma P^\pi]^{-1} \tag{28}$$

and

$$\frac{\partial P^\pi}{\partial \pi_j(x', y')} = \begin{cases} 0 & \text{if } x \neq x' \\ P_j^{\pi_{-j}}(s', x, y) & \text{else.} \end{cases} \tag{29}$$

Clearly, this requires inverting the matrix $[I - \gamma P^\pi]$. Note that $P^\pi$ is a square state transition matrix with distributions on columns. By the same argument as Lemma 1, this matrix has full-row rank, and since it is square, it is non-singular, and hence invertible. Therefore, the derivative (27) always exists. $\qquad\square$

## C. KKT Conditions Imply Fixed Point Sufficiency

Consider the following constrained optimization problem:

$$\max_{\boldsymbol{x} \in \mathbb{R}^d} f(\boldsymbol{x}) \tag{30a}$$

$$s.t. \; g_i(\boldsymbol{x}) \leq 0 \; \forall i \tag{30b}$$

$$h_j(\boldsymbol{x}) = 0 \; \forall j \tag{30c}$$

where $f$ is concave and $g_i$ and $h_j$ represent inequality and equality constraints respectively. If $g_i$ and $h_i$ are affine functions, then any maximizer $\boldsymbol{x}^*$ of $f$ must satisfy the following necessary and sufficient KKT conditions (Ghojogh et al., 2021; Boyd & Vandenberghe, 2004):

- Stationarity: $\mathbf{0} \in \partial f(\boldsymbol{x}^*) - \sum_j \lambda_j \partial h_j(\boldsymbol{x}^*) - \sum_i \mu_i \partial g_i(\boldsymbol{x}^*)$

- Primal feasibility: $h_j(\boldsymbol{x}^*) = 0$ for all $j$ and $g_i(\boldsymbol{x}^*) \leq 0$ for all $i$

- Dual feasibility: $\mu_i \geq 0$ for all $i$

- Complementary slackness: $\mu_i g_i(\boldsymbol{x}^*) = 0$ for all $i$.

**Lemma 3.** *Assuming player $k$'s utility, $u_k(x_k, x_{-k})$, is concave in its own strategy $x_k$, a strictly-positive primal-feasible strategy is a best response $BR_k$ if and only if it has zero projected-gradient norm.*

*Proof.* Consider the problem of formally computing $\epsilon_k(\boldsymbol{x}) = \max_{z \geq 0, Az = b} u_k(z, x_{-k}) - u_k(x_k, x_{-k})$:

$$\max_{z \in \mathbb{R}^d} u_k(z, x_{-k}) - u_k(x_k, x_{-k}) \tag{31a}$$

$$s.t. -z_k \leq 0 \ \forall k \tag{31b}$$

$$A_j z - b_j = 0 \ \forall j. \tag{31c}$$

Note that the objective is linear (concave) in $z$ and the constraints are affine, therefore the KKT conditions are necessary and sufficient for optimality. Recall that we assume that the solution $z^*$ is positive, $z_k^* > 0$ for each $k$. Also, let $e_k$ be a onehot vector, i.e., a zeros vector except with a 1 at index $k$. Mapping the KKT conditions onto this problem yields the following:

- Stationarity: $\mathbf{0} \in \partial u_k(z^*, x_{-k}) - \sum_j \lambda_j A_j + \sum_k \mu_k e_k$

- Primal feasibility: $A_j z = b_j$ for all $j$

- Dual feasibility: $\mu_i \geq 0$ for all $k$

- Complementary slackness: $-\mu_k z_k^* = 0$ for all $k$

where $\partial u_k(z, \cdot)$ is the subdifferential at $z$. Consider any primal-feasible point $Az^* = b$. Given our assumption that $z_k^* > 0$, by complementary slackness and dual feasibility, each $\mu_k$ must be identically zero. This implies the stationarity condition can be simplified to $\mathbf{0} \in \partial u_k(z^*, x_{-k}) - \sum_j \lambda_j A_j = \partial u_k(z^*, x_{-k}) - A^\top \lambda$. Rearranging terms we find that for any $z^*$, there exists $\lambda$ such that

$$\sum_j \lambda_j A_j \in \partial u_k(z^*, x_{-k}). \tag{32}$$

Equivalently, elements of $\partial u_k(z^*, x_{-k})$ are in the row-span of $A$.

For the rest of the proof, we follow the derivation of the gradient projection method (Luenberger et al., 1984, Sec 12.4, p 364). Let $\nabla_k \in \partial u_k$ be a subderivative (gradient), i.e., an element of the subdifferential.

Note that, in general, we can write any gradient as a sum of elements from the row-span of $A$ and its orthogonal complement $d_k$, which lies in the tangent space of the feasible set:

$$\nabla_k = d_k + A^\top \lambda. \tag{33}$$

We may solve for $\lambda$ through the requirement that $Ad_k = 0$, i.e., any movement within the tangent space of the feasible set remains feasible. Thus

$$Ad_k = A\nabla_k - (AA^\top)\lambda = 0 \tag{34}$$

$$\implies \lambda = (AA^\top)^{-1} A\nabla_k \tag{35}$$

$$\implies d_k = \nabla_k - A^\top \lambda = [I - A^\top(AA^\top)^{-1}A]\nabla_k \tag{36}$$

$$= \Pi_{TA}(\nabla_k) \tag{37}$$

where $\Pi_{TA}[I - A^\top(AA^\top)^{-1}A]$ is the matrix that projects any gradient vector onto the tangent space of the feasible set given by the constraint matrix $A$.

The fact that elements of $\partial u_k(z^*, x_{-k})$ are in the row-span of $A$ implies that $0 = d_k = \Pi_{TA}(\nabla_k)$ necessarily, completing the claim. $\square$

## D. Entropy Regularized Loss Bounds Exploitability

Our derived exploitability bound only requires concavity of the utility, bounded diameter of the feasible set, and linearity of feasible constraints (i.e., feasible set is subset of hyperplane). In what follow, let player $k$'s loss be the negative of their utility, i.e., $\ell_k = -u_k$.

**Lemma 4.** *The amount a player can gain by deviating is upper bounded by a quantity proportional to the norm of the projected-gradient:*

$$\epsilon_k(\boldsymbol{\mu}) \leq \sqrt{2}||\Pi_{T\mathcal{U}_k}(\nabla_{\mu_k}^k)||. \tag{38}$$

*Proof.* Let $z$ be any point in the feasible set. First note that $\Pi_{T\mathcal{U}_k}(z) = Bz$ where $B$ is an orthogonal projection matrix; this implies $B^2 = B = B^\top$. Then by convexity of $\ell_k$ with respect to $z$,

$$\ell_k(\boldsymbol{\mu}) - \ell_k(z, \mu_{-k}) \leq (\nabla_{\mu_k}^k)^\top(\mu_k - z) \tag{39a}$$

$$= (\nabla_{\mu_k}^k)^\top \Pi_{T\mathcal{U}_k} \underbrace{(\mu_k - z)}_{\in T\mathcal{U}} \tag{39b}$$

$$= \underbrace{(\Pi_{T\mathcal{U}_k}(\nabla_{\mu_k}^k))^\top}_{Bz = B^\top z} \underbrace{(\mu_k - z)}_{\text{Diam}(\mathcal{M}_k) \leq \sqrt{2}} \tag{39c}$$

$$\leq \sqrt{2}||\Pi_{T\mathcal{U}_k}(\nabla_{\mu_k}^k)|| \tag{39d}$$

where the first equality follows from the fact any two points $z$ and $\mu_k$ lying in the same hyperplane, by definition, form a direction lying in the tangent space of the hyperplane. The second equality follows from symmetry of the projection operator and simply grouping its application to the left hand term; we also note that the feasible set is a subset of the simplex, which has a diameter of $\sqrt{2}$. Finally, the last step follows from Cauchy-Schwarz. $\square$

**Theorem 2** (Low Temperature Approximate Equilibria are Approximate Nash Equilibria). *Let $\nabla_{\mu_k}^{k\tau}$ be player $k$'s entropy regularized gradient and $\boldsymbol{\mu}$ be an approximate equilibrium of the entropy regularized game. Then it holds that*

$$\epsilon_k = \ell_k(\boldsymbol{\mu}) - \ell_k(BR_k, \mu_{-k}) \leq \tau \log(|\mathcal{S}||\mathcal{A}_k|) + \sqrt{2}||\Pi_{T\mathcal{U}_k}(\nabla_{\mu_k}^{k\tau})||. \tag{40}$$

*Proof.* Beginning with the definition of exploitability, we find

$$\ell_k(\boldsymbol{\mu}) - \ell_k(BR_k, \mu_{-k}) = \big(\ell_k(\boldsymbol{x}) + \tau S(\mu_k) - \tau S(\mu_k)\big) \tag{41a}$$

$$- \big(\ell_k(BR_k, \mu_{-k}) + \tau S(BR_k) - \tau S(BR_k)\big)$$

$$= \ell_k^\tau(\boldsymbol{\mu}) - \ell_k^\tau(BR_k, \mu_{-k}) + \tau\big(S(\mu_k) - S(BR_k)\big) \tag{41b}$$

$$\leq \ell_k^\tau(\boldsymbol{\mu}) - \min_{z \in \mathcal{M}_k} \ell_k^\tau(z, \mu_{-k}) + \tau \max_{z' \in \mathcal{M}_k} S(z') \tag{41c}$$

$$\leq \sqrt{2}||\Pi_{T\mathcal{U}_k}(\nabla_{\mu_k}^{k\tau})|| + \tau \max_{z' \in \mathcal{M}_k} S(z') \tag{41d}$$

$$\leq \sqrt{2}||\Pi_{T\mathcal{U}_k}(\nabla_{\mu_k}^{k\tau})|| + \tau \log(|\mathcal{S}||\mathcal{A}_k|) \tag{41e}$$

where the second equality follows from the definition of player $k$'s entropy regularized loss $\ell_k^\tau$, the first inequality from nonnegativity of entropy $S$, the second inequality from convexity of $\ell_k^\tau$ with respect to its first argument (Lemma 4), and the last from the maximum possible value of Shannon entropy over distributions on $|\mathcal{S}||\mathcal{A}_k|$ elements. $\square$

**Corollary 1** ($\mathcal{L}^\tau$ Scores Nash Equilibria). *Let $\mathcal{L}^\tau(\boldsymbol{\mu})$ be our proposed entropy regularized loss function and $\boldsymbol{\mu}$ be any strategy profile. Then it holds that*

$$\epsilon \leq \tau \log(|\mathcal{S}| \max_k |\mathcal{A}_k|) + \sqrt{2n\mathcal{L}^\tau(\boldsymbol{\mu})}. \tag{42}$$

*Proof.* Beginning with the definition of exploitability and applying Lemma 2, we find

$$\epsilon = \max_k \max_z \ell_k(\boldsymbol{\mu}) - \ell_k(z, \mu_{-k}) \qquad \text{(recall each } \epsilon_k \geq 0) \tag{43a}$$

$$\leq \max_k \left[ \tau \log(|\mathcal{S}||\mathcal{A}_k|) + \sqrt{2} ||\Pi_{T\mathcal{U}_k}(\nabla_{\mu_k}^{k\tau})|| \right] \tag{43b}$$

$$\leq \tau \max_k \log(|\mathcal{S}||\mathcal{A}_k|) + \sqrt{2} \max_k ||\Pi_{T\mathcal{U}_k}(\nabla_{\mu_k}^{k\tau})|| \tag{43c}$$

$$\leq \tau \max_k \log(|\mathcal{S}||\mathcal{A}_k|) + \sqrt{2} \sum_k ||\Pi_{T\mathcal{U}_k}(\nabla_{\mu_k}^{k\tau})|| \tag{43d}$$

$$= \tau \log(|\mathcal{S}| \max_k |\mathcal{A}_k|) + \sqrt{2} \left| \left| ||\Pi_{T\mathcal{U}_1}(\nabla_{\mu_1}^{1\tau})||_2, \ldots, ||\Pi_{T\mathcal{U}_n}(\nabla_{\mu_n}^{n\tau})||_2 \right| \right|_1 \tag{43e}$$

$$\leq \tau \log(|\mathcal{S}| \max_k |\mathcal{A}_k|) + \sqrt{2n} \left| \left| ||\Pi_{T\mathcal{U}_1}(\nabla_{\mu_1}^{1\tau})||_2, \ldots, ||\Pi_{T\mathcal{U}_n}(\nabla_{\mu_n}^{n\tau})||_2 \right| \right|_2 \tag{43f}$$

$$\leq \tau \log(|\mathcal{S}| \max_k |\mathcal{A}_k|) + \sqrt{2n} \sqrt{\sum_k ||\Pi_{T\mathcal{U}_k}(\nabla_{\mu_k}^{k\tau})||_2^2} \tag{43g}$$

$$= \tau \log(|\mathcal{S}| \max_k |\mathcal{A}_k|) + \sqrt{2n\mathcal{L}^\tau(\boldsymbol{\mu})}. \tag{43h}$$

$$\square$$

# E. PGL Hyperparameters

Table 5 lists out the three types of annealing schedules used in the experiments. Table 6 lists out the learning rate used in each domain as well as the type of annealing schedule used. All experiments were run with Adam (Kingma & Ba, 2015).

| Requirements to Anneal | | |
|---|---|---|
| Minimum Temperature | $\tau_t \leftarrow \max(\tau_t, \min \tau)$ | $10^{-2}$ |
| Minimum Iterations Per Temperature | $\min_{\Delta t}\{\tau_t - \tau_{t+\Delta t}|\tau_t - \tau_{t+\Delta t} > 0\}$ | 50 |
| Loss Threshold Requirement for Annealing | $\mathcal{L}^{\tau_t} \leq \epsilon$ | $10^{-1}$ |
| Anneal Rule (When Requirements Met) | | |
| Type 1 Anneal Rate | $\tau_t$ | $\tau = 10^{-\lfloor t/1000 \rfloor}$ |
| Type 2 Anneal Rate | $\tau_{t+1}/\tau_t$ | 0.8 |
| Type 3 Anneal Rule | $\tau_{t+1}$ | $\tau_t + \frac{1}{10} \cdot \mathcal{L}^{\tau_t} / \min(\frac{\partial \mathcal{L}^\tau}{\partial \tau}, -\mathcal{L}^\tau)$ |

*Table 5.* Annealing Schedules

| Domain / Hyperparameter | Learning Rate | Anneal Type | Iterations ($T$) |
|---|---|---|---|
| IPD (Creativity) | $10^{-1}$ | Type 1 | $8 \times 10^3$ |
| IPD (Imitation) | $10^{-2}$ | Type 1 | $8 \times 10^3$ |
| IPGG | $10^{-1}$ | Type 1 | $8 \times 10^3$ |
| El Farol | $10^{-1}$ | Type 1 | $8 \times 10^3$ |
| Bach-Stravnisky | $10^{-1}$ | Type 1 | $1 \times 10^3$ |
| Synthetic | $10^{-1}$ | Type 2 | $8 \times 10^3$ |
| Robot Warehouse | $10^{-2}$ | Type 2 | $8 \times 10^3$ |
| Pathfinding | $10^{-2}$ | Type 3 | $1 \times 10^6$ |

*Table 6.* Learning Rate and Annealing Schedule by Domain.

# F. Descriptions of Domains

Here, we provide further details on the domains we used in experiments.

## F.1. Iterated Normal-form Games

We provide the payoffs for the iterated prisoner's dilemma and Bach-Stravinsky game used in experiments. See section 5 in the main body for a description of the iterated public goods game and El Farol bar problem.

|   | $C$ | $D$ |
|---|-----|-----|
| $C$ | $-1, -1$ | $-3, 0$ |
| $D$ | $0, -3$ | $-2, -2$ |

$\implies$

|   | $C$ | $D$ |
|---|-----|-----|
| $C$ | $2/3, 2/3$ | $0, 1$ |
| $D$ | $1, 0$ | $1/3, 1/3$ |

*Figure 4.* Prisoner's Dilemma Game: We shift and normalize the payoffs of the prisoner's dilemma game (left) to be non-negative and with maximum value 1 (right).

|   | $C$ | $D$ |
|---|-----|-----|
| $C$ | $3, 2$ | $0, 0$ |
| $D$ | $0, 0$ | $2, 3$ |

*Figure 5.* Bach-Stravinsky Game.

### F.2. Grid World Domains

**Pathfinding**  We use OpenSpiel's (Lanctot et al., 2019) pathfinding game with a horizon of 1000: https://openspiel.readthedocs.io/en/latest/games.html. See Figure 2 for a visual of the grid specification. To enable the game with an infinite horizon, we transition all agents back to their starting positions (left most grid in Figure 2) after they reach the goal state. All positive rewards in OpenSpiel's variant are replaced with $+100$.

**Synthetic Illustrative Safety Domain**  We consider a simple 2-player, 2-state, 2-action, symmetric convex Markov game. If both agents select action $0$, they transition from their current state to the other state; otherwise, they remain put. We set $\gamma = 0.95$ and the initial state measure $\mu_0$ to be uniform.

For this domain, we pose a "safe" MARL objective; we designate a set of safe long-run visitation measures for certain states and usage of certain actions; equivalently, we rule out certain unsafe states and actions. As an example of "safe" MARL, define the following convex loss (negative utility) over occupancy measures for player 1:

$$-u_i(\mu_i) = \ell_i(\mu_i) = \max(0, ||\mu_a - t_a||_\infty - 1/20) + \max(0, ||\mu_s - t_s||_\infty - 1/4) \tag{44}$$

where $\mu_a$ and $\mu_s$ are player 1's action and state marginals respectively. The target measures $t_a$ and $t_s$ are used along with the radii $1/20$ and $1/4$ to encode the regions of "safe" occupancy measures. If player 1 deviates from either region by more than the radius (as measured by the infinity norm), then they accrue a loss. Otherwise, player 1's loss is zero.

In Figure 1, we fix player 2 to use the following policy:

$$\pi_2^I(a_0|s) = \begin{cases} 0.40 & \text{if } s = s_0 \\ 0.80 & \text{else} \end{cases},$$

We provide numpy code describing the transition kernel $P$ and reward function $r(s, a)$ (denoted by `pt` for "payoff tensor" below) for the synthetic safety domain used in Figure 1.

```
ns = 2
npl = 2
na = np.ones(npl, dtype=int) * 2

epsilon = 0.0
transition_env = np.zeros((ns, ns) + tuple(na))

# a=0 (coordinate), a=1 (not coordinate)
transition_env[1, 0, 0, 0] = 1 - epsilon
transition_env[1, 0, 0, 1] = epsilon
transition_env[1, 0, 1, 0] = epsilon
transition_env[1, 0, 1, 1] = 0

transition_env[0, 1, 0, 0] = 1 - epsilon
```

```
transition_env[0, 1, 0, 1] = epsilon
transition_env[0, 1, 1, 0] = epsilon
transition_env[0, 1, 1, 1] = 0

# only two states so prob of one is 1 minus prob of other
transition_env[0, 0, :, :] = 1 - transition_env[1, 0, :, :]
transition_env[1, 1, :, :] = 1 - transition_env[0, 1, :, :]

assert np.all(np.sum(transition_env, axis=0) == 1)

self.transition_env = transition_env.astype(float)
self.pt = np.zeros((npl, ns) + tuple(na), dtype=float)
```

**Robot Warehouse Safety Grid World Domain**    We provide numpy code describing the transition kernel $P$ and reward function $r(s, a)$ (denoted by `pt` for "payoff tensor" below) for the robot warehouse safety domain described in section 5.

This domain represents a 3-cell grid world where two robots pick up and drop off packages (Left Drop-off ↔ Package Pickup ↔ Right Drop-off). The pickup point is the middle cell and the two robots drop off in the cells on either side. The left robot always drops off on the left; the right always drops off on the right. They can move slow or fast (2 actions each). They get +1 for slowly dropping off package, +2 for quickly dropping of a package. They always alternate between picking up and dropping off, but their chosen speed affects their probability of moving from the pickup to the drop-off point. Two robots attempting fast pickups is dangerous, so we construct a convex loss to penalize that behavior. At the same time, the robots are incentivized to move fast at the pickup point (to move to the drop-off and earn more reward). The domain exhibits an additional complexity given by the following social dilemma. Moving fast at the pickup point when the other is moving slowly increases the probability of successfully picking up a package and moving to the drop off; moving slow with a fast partner results in much lower probability. This simulates a scenario where packages arrive at the drop-off at a constant rate.

```
ns = 4
npl = 2
na = np.ones(npl, dtype=int) * 2

transition_env = np.zeros((ns, ns) + tuple(na))

# s=0: (pickup, pickup)
# s=1: (pickup, dropoff)
# s=2: (dropoff, pickup)
# s=3: (dropoff, dropoff)

# probability of moving from dropoff to pickup is independent of the other
# agent
p_reset = 1.0
# probability of moving from pickup to dropoff is independent of other agent
# if alone at pickup point
p_drop_alone_slow = 0.7
p_drop_alone_fast = 0.8
# generic low, medium, and high probabilities
p_low = 0.2
p_mid = 0.5
p_high = 0.8

# a=0 (slow), a=1 (fast)

# reward is +1 for visiting dropoff state, else 0
reward = np.zeros((npl, ns,) + tuple(na))
```

```
# reward[1, 1] = 1.0
# reward[0, 2] = 1.0
# reward[:, 3] = 1.0
reward[1, 1, :, 0] = 1.0
reward[1, 1, :, 1] = 2.0
reward[0, 2, 0, :] = 1.0
reward[0, 2, 1, :] = 2.0
reward[0, 3, 0, :] = 1.0
reward[0, 3, 1, :] = 2.0
reward[1, 3, :, 0] = 1.0
reward[1, 3, :, 1] = 2.0

#### s'=0 #####
# s=0 --> s'=0: (pickup, pickup) --> (pickup, pickup)
# define s'= 1, 2, 3 first
# skip
# transition_env[0, 0, 0, 0] =
# transition_env[0, 0, 0, 1] =
# transition_env[0, 0, 1, 0] =
# transition_env[0, 0, 1, 1] =

# s=1 --> s'=0: (pickup, dropoff) --> (pickup, pickup)
transition_env[0, 1, 0, 0] = (1.0 - p_drop_alone_slow) * p_reset
transition_env[0, 1, 0, 1] = (1.0 - p_drop_alone_slow) * p_reset
transition_env[0, 1, 1, 0] = (1.0 - p_drop_alone_fast) * p_reset
transition_env[0, 1, 1, 1] = (1.0 - p_drop_alone_fast) * p_reset

# s=2 --> s'=0: (dropoff, pickup) --> (pickup, pickup)
transition_env[0, 2, 0, 0] = p_reset * (1.0 - p_drop_alone_slow)
transition_env[0, 2, 0, 1] = p_reset * (1.0 - p_drop_alone_fast)
transition_env[0, 2, 1, 0] = p_reset * (1.0 - p_drop_alone_slow)
transition_env[0, 2, 1, 1] = p_reset * (1.0 - p_drop_alone_fast)

# s=3 --> s'=0: (dropoff, dropoff) --> (pickup, pickup)
transition_env[0, 3, 0, 0] = p_reset
transition_env[0, 3, 0, 1] = p_reset
transition_env[0, 3, 1, 0] = p_reset
transition_env[0, 3, 1, 1] = p_reset

#### s'=1 #####
# s=0 --> s'=1: (pickup, pickup) --> (pickup, dropoff)
transition_env[1, 0, 0, 0] = (1 - p_mid) * p_mid
transition_env[1, 0, 0, 1] = p_high
transition_env[1, 0, 1, 0] = p_low
transition_env[1, 0, 1, 1] = p_low

# s=1 --> s'=1: (pickup, dropoff) --> (pickup, dropoff)
transition_env[1, 1, 0, 0] = (1 - p_drop_alone_slow) * (1.0 - p_reset)
transition_env[1, 1, 0, 1] = (1 - p_drop_alone_slow) * (1.0 - p_reset)
transition_env[1, 1, 1, 0] = (1 - p_drop_alone_fast) * (1.0 - p_reset)
transition_env[1, 1, 1, 1] = (1 - p_drop_alone_fast) * (1.0 - p_reset)

# s=2 --> s'=1: (dropoff, pickup) --> (pickup, dropoff)
transition_env[1, 2, 0, 0] = p_reset * p_drop_alone_slow
```

```
transition_env[1, 2, 0, 1] = p_reset * p_drop_alone_fast
transition_env[1, 2, 1, 0] = p_reset * p_drop_alone_slow
transition_env[1, 2, 1, 1] = p_reset * p_drop_alone_fast

# s=3 --> s'=1: (dropoff, dropoff) --> (pickup, dropoff)
transition_env[1, 3, 0, 0] = p_reset * (1.0 - p_reset)
transition_env[1, 3, 0, 1] = p_reset * (1.0 - p_reset)
transition_env[1, 3, 1, 0] = p_reset * (1.0 - p_reset)
transition_env[1, 3, 1, 1] = p_reset * (1.0 - p_reset)

#### s'=2 #####
# s=0 --> s'=2: (pickup, pickup) --> (dropoff, pickup)
transition_env[2, 0, 0, 0] = p_mid * (1 - p_mid)
transition_env[2, 0, 0, 1] = p_low
transition_env[2, 0, 1, 0] = p_high
transition_env[2, 0, 1, 1] = p_low

# s=1 --> s'=2: (pickup, dropoff) --> (dropoff, pickup)
transition_env[2, 1, 0, 0] = p_drop_alone_slow * p_reset
transition_env[2, 1, 0, 1] = p_drop_alone_slow * p_reset
transition_env[2, 1, 1, 0] = p_drop_alone_fast * p_reset
transition_env[2, 1, 1, 1] = p_drop_alone_fast * p_reset

# s=2 --> s'=2: (dropoff, pickup) --> (dropoff, pickup)
transition_env[2, 2, 0, 0] = (1.0 - p_reset) * (1.0 - p_drop_alone_slow)
transition_env[2, 2, 0, 1] = (1.0 - p_reset) * (1.0 - p_drop_alone_fast)
transition_env[2, 2, 1, 0] = (1.0 - p_reset) * (1.0 - p_drop_alone_slow)
transition_env[2, 2, 1, 1] = (1.0 - p_reset) * (1.0 - p_drop_alone_fast)

# s=3 --> s'=2: (dropoff, dropoff) --> (dropoff, pickup)
transition_env[2, 3, 0, 0] = (1.0 - p_reset) * p_reset
transition_env[2, 3, 0, 1] = (1.0 - p_reset) * p_reset
transition_env[2, 3, 1, 0] = (1.0 - p_reset) * p_reset
transition_env[2, 3, 1, 1] = (1.0 - p_reset) * p_reset

#### s'=3 #####
# s=0 --> s'=3: (pickup, pickup) --> (dropoff, dropoff)
transition_env[3, 0, 0, 0] = p_mid
transition_env[3, 0, 0, 1] = p_low
transition_env[3, 0, 1, 0] = p_low
transition_env[3, 0, 1, 1] = p_low

# s=1 --> s'=3: (pickup, dropoff) --> (dropoff, dropoff)
transition_env[3, 1, 0, 0] = 1.0 - transition_env[:, 1, 0, 0].sum()
transition_env[3, 1, 0, 1] = 1.0 - transition_env[:, 1, 0, 1].sum()
transition_env[3, 1, 1, 0] = 1.0 - transition_env[:, 1, 1, 0].sum()
transition_env[3, 1, 1, 1] = 1.0 - transition_env[:, 1, 1, 1].sum()

# s=2 --> s'=3: (dropoff, pickup) --> (dropoff, dropoff)
transition_env[3, 2, 0, 0] = 1.0 - transition_env[:, 2, 0, 0].sum()
transition_env[3, 2, 0, 1] = 1.0 - transition_env[:, 2, 0, 1].sum()
transition_env[3, 2, 1, 0] = 1.0 - transition_env[:, 2, 1, 0].sum()
transition_env[3, 2, 1, 1] = 1.0 - transition_env[:, 2, 1, 1].sum()
```

```
# s=3 --> s'=3: (dropoff, dropoff) --> (dropoff, dropoff)
transition_env[3, 3, 0, 0] = 1.0 - transition_env[:, 3, 0, 0].sum()
transition_env[3, 3, 0, 1] = 1.0 - transition_env[:, 3, 0, 1].sum()
transition_env[3, 3, 1, 0] = 1.0 - transition_env[:, 3, 1, 0].sum()
transition_env[3, 3, 1, 1] = 1.0 - transition_env[:, 3, 1, 1].sum()

#### s'=0 #####
# s=0 --> s'=0: (pickup, pickup) --> (pickup, pickup)
transition_env[0, 0, 0, 0] = 1.0 - transition_env[:, 0, 0, 0].sum()
transition_env[0, 0, 0, 1] = 1.0 - transition_env[:, 0, 0, 1].sum()
transition_env[0, 0, 1, 0] = 1.0 - transition_env[:, 0, 1, 0].sum()
transition_env[0, 0, 1, 1] = 1.0 - transition_env[:, 0, 1, 1].sum()

assert np.allclose(np.sum(transition_env, axis=0), 1.0)

self.transition_env = transition_env.astype(float)
self.pt = reward.astype(float)
```

