# OpenReview forum: "Convex Markov Games: A New Frontier for Multi-Agent Reinforcement Learning"
_ICML.cc/2025/Conference — ICML 2025 poster_

### Official Review · Reviewer_9J7p · 2025-03-11

**Overall Recommendation:** 5

**Summary:**

The paper proposes a new model for multi-agent interaction called Convex Markov Games (CMGs), that generalizes the concept of Markov Games to convex objectives of the induced state distribution. The authors characterise the existence of mixed and pure Nash Equilibria and propose a simple algorithm to find them. Finally, they show how such an algorithm can be used to enforce non-trivial behaviours on many relevant instances subsumed by cMGs.

## Update after rebuttal

I went through the authors' responses and other reviews briefly, the authors successfully addressed my main concerns regarding the work and more crucially reviewer Bm64's concerns, so I would recommend accepting the paper.

**Claims And Evidence:**

Yes, the proofs are rigorous and the empirical evidence is convincing.

**Essential References Not Discussed:**

No

**Experimental Designs Or Analyses:**

Yes, all the ones in Section 5. They are sound and show valid results.

**Methods And Evaluation Criteria:**

The paper is more of an exploratory work, but the empirical instantiations of the convex utility are convincing and relevant.

**Other Comments Or Suggestions:**

- I would reference the proof of Eq 12 in Appendix (Corollary 1)
- I would anticipate the description of the Projection operator after Theorem 2.
- I would describe the properties (and limitations of Algorithm 1 as described in the conclusions) more explicit.

**Other Strengths And Weaknesses:**

Strenghts:
- The paper is extremely well-written and rigorous in the exposition as well.
- The decision-making problem is relevant and the empirical corroboration is rather interesting.

Weaknesses:
- The theoretical contributions are mostly built on a combination of previous results, this is far from being an issue per se, as they are applied to novel problems, but I would make this fact slightly more explicit in the main paper as well, like directly citing references in theorem statements.

**Questions For Authors:**

- Is the main outcome of the paper that convex MGs are structurally not to diverse from their non-convex counterpart? As most of results for linear games do transfer. In case, I would make this statement more clear in the introduction.

**Relation To Broader Scientific Literature:**

This work is a fundamental contribution for the convex generalisation of Markov Games and their characterisation. It shows that common arguments can be applied to these games as well (results in Prop.1, Th.1, Th.2) with success.

**Theoretical Claims:**

Yes. Prop. 1, Th. 1, Th.2 have been checked, I haven't found any issue with them.

---

> ### Author Rebuttal · Authors · 2025-04-01
>
> Dear reviewer, thank you for your positive endorsement and helpful comments. We are pleased to hear you not only appreciate our proposed generalization of Markov games, but also checked and found **“the proofs are rigorous and the empirical evidence is convincing”**.  Thank you for saying the **“paper is extremely well-written and rigorous in the exposition”**; we will further improve the writing with your comments with proposed changes listed below.
>
> - We can add a comment that Prop 1, Th 1, and Th 2 are novel applications of Glicksberg, Debreau/Kosowsky, and Gemp to the cMG framework.
> - We will reference Corollary 1 as you say.
> - We can move the discussion around (13) to be after Theorem 2.
> - We can move/copy the last paragraph of the conclusion to a limitations paragraph immediately after the Algorithm 1 code block. We can also mention some impossibility results on computing NEs and point out that PGL is not guaranteed to converge to NEs.
>
> Regarding the main outcome of the paper, we are not sure we want to make a general statement that cMGs are structurally similar to MGs. It is true that pure NEs still exist, but value functions do not. And at least the reason pure NE exists requires more sophisticated and different proof techniques while prior existence proofs leveraged the existence of value functions. However, there is a key point of similarity between them. cMGs (similar to MGs) admit a clear solution concept that enables a formal computational investigation, which hopefully similarly to the case of MGs will allow for many theoretical and practical research advances.

---

> > ### Comment · Reviewer_9J7p · 2025-04-02
> >
> > Given the Rebuttal, I am positive with confirming my score, provided that the authors address the clarity issues listed by the Reviewers and include the relevant missing literature outlined by Bm64. Additionally, I would strongly suggest to include a reference to the concurrent [1], which introduces a definition of convex Markov Games as well, even though with a slightly different definition and with a different focus in contributions.
> >
> > [1] Zamboni et al., Towards Principled Multi-Agent Task Agnostic Exploration, 2025

---

### Official Review · Reviewer_Bm64 · 2025-03-12

**Overall Recommendation:** 4

**Summary:**

The paper presents convex Markov games, a framework that extends Markov games by generalizing the players' linear utilities with any convex function of their state-action occupancy measures. A similar generalization recently studied in single-agent problems, from MDP to convex MDPs. Here the same extension is done for multi-agent problems. First, the paper states the existence of mixed-strategy Nash equilibria in convex Markov games. Then, the paper presents a method to compute approximate equilibria in practice. The method is based on projected gradient descent of an upper bound to the exploitability of each agent policy, which is in turn related to Nash equilibria. The method is evaluated in a variety of small-scale domains with knowledge of the transition parameters, comparing the proposed algorithm with various (self-designed) baselines, such as simultaneous gradient descent and black-box exploitability minimization.

**Claims And Evidence:**

There is a potential error in the way pure/mixed strategies are defined. The paper says "If all probability mass of $\rho$ is on a single policy $\pi_i$ [also stochastic], we call $\rho$ a pure-strategy and write $\pi_i$ directly". I am not sure this makes sense, because the occupancy measure of any stochastic policy can be equivalently seen as the occupancy measure of a mixture of policies. The policies in the mixture do not even need to be stochastic. This is a classical result that can be found in (Puterman 2014). If this is correct and I am not missing something, it would mean that mixed strategy and pure strategy are the same thing here. Hopefully, this is not too problematic, but it shall be fixed (or explained).

----

Solved below: Increasing my score to accept

**Essential References Not Discussed:**

The account of prior works in convex/general utilities RL is very limited. A few related references are missing, I am providing a (possibly incomplete) list below:
- Cheung, Regret minimization for reinforcement learning with vectorial feedback and complex objectives, 2019
- Cheung, Exploration-exploitation trade-off in reinforcement learning on online Markov decision processes with global concave rewards, 2019
- Geist et al., Concave utility reinforcement learning: The mean-field game viewpoint, 2021
- Mutti et al., Convex reinforcement learning in finite trials, 2023
- Barakat et al., Reinforcement Learning with General Utilities: Simpler Variance Reduction and Large State-Action Space, 2023
- Moreno et al., Efficient model-based concave utility reinforcement learning through greedy mirror descent, 2024
- Moreno et al., MetaCURL: Non-stationary Concave Utility Reinforcement Learning, 2024
- Prajapat et al., Submodular reinforcement learning, 2024
- Celikok et al., Inverse Concave-Utility Reinforcement Learning is Inverse Game Theory, 2024
- De Santi et al., Global Reinforcement Learning: Beyond Linear and Convex Rewards via Submodular Semi-gradient Methods, 2024

I think those paper shall at least be mentioned in the manuscript and, in some cases, discussed in details.

**Experimental Designs Or Analyses:**

The experimental analysis is carried out in small-scale domains with knowledge of the transition model. It is fine at this stage of development, but perhaps mentioning multi-agent RL in the title can be misleading on the kind of challenges addressed in the paper.

**Methods And Evaluation Criteria:**

Since this is a new framework, there aren't obvious baselines to compare the proposed algorithm to. However, this makes the significance of the algorithmic contribution unclear. Is it problematic to adapt any other algorithm that works for convex MDPs to the multi-agent version? There is not much convincing discussion on that.

**Other Comments Or Suggestions:**

The introduction could place more clear credit to convex MDP literature. To the best of my knowledge, the problem has been introduced by Hazan et al (2019) and developed in several subsequent works. An overview of previous results could help clarify the unique challenges of the multi-agent setting. Same goes for the prior works and results on Markov games.

Moreover, the literature of convex RL includes various alternative formulations of the problem, such as convex RL (Hazan et al. 2019), finite-trial convex RL (Mutti et al. 2023), submodular RL (Prajapat et al. 2023). Perhaps mentioning why a formulation has been preferred to another for the extension to Markov games could be useful.

The introduction could also give more intuitive explanations on how a convex MG works, which is only introduced in the subsequent section.

**Other Strengths And Weaknesses:**

Strengths
- Natural extension of prior work in single-agent convex utilities decision making;
- There seems to be several interesting applications fitting the framework;
- The algorithm is promising, although it lacks intuition and clear motivation.

Weaknesses
- An approximate algorithm is given before even discussing the computational complexity of the original problem;
- Which guarantees does the algorithm have in general? Can we give stronger guarantees under common game-theoretic assumptions (e.g, zero sum, potential games...)?
- Issue with the definition of mixed/pure strategies mentioned above;
- Limited experimental analysis, without competitive baselines and RL.

This paper addresses a very nice direction, which may have potential impact on the understanding of decision making beyond linear utilities, building on prior results on single-agent settings. I am currently providing a borderline evaluation given my confusion over the connection between the results reported in the paper and related works on Markov games and convex MDPs. However, the framework may be an important contribution in itself and I may be swayed towards acceptance with a convincing author response.

**Questions For Authors:**

What is the complexity of computing equilibria in this setting? It appears that the problem is intractable even for general-sum Markov games. I guess it is not tractable for general convex as well.

Projection over the tangent space of $\mathcal{U}_i$: Can you provide some intuition on the meaning of the projection? Why is this necessary, beyond the link with exploitability upper bound?

The experiments are a little hard to process: How do we know that PGL is a good algorithm for cMG?

Is the symbol $r_i$ introduced somewhere? I guess it is a reward: Why do we need it in the first place?

Is PGL guaranteed to converge to an approximate equilibrium?

The title mention RL, but there's not much RL in the paper. How can ideas be translated to large problems or/and unknown transitions?

**Relation To Broader Scientific Literature:**

This looks like a very natural extension of the recent stream of works on convex utilities in RL. In terms of results, I feel like a tighter connections with the theory of Markov games could be drawn here. I am not an expert of the latter literature, but it is definitely hard to grasp the additional computational/statistical challenges introduced by convex utilities w.r.t. what is known in Markov games.

**Theoretical Claims:**

I didn't carefully check the correctness of the proofs and the theory beyond what is included in the main paper. I reported a potential issue above.

---

> ### Author Rebuttal · Authors · 2025-04-01
>
> Dear reviewer, thanks for your comments and highlighting our work as a **“Natural extension of prior work in single-agent convex utilities decision making”** with **“several interesting applications”**. Your feedback will help us greatly improve the paper.
>
> Pure/Mixed Strategies and Deterministic/Stochastic Policies: Thank you for raising this point of confusion. Reading your comment, we agree factually with each technical statement, but are confused by your inference. Recall that we define strategies in terms of policies, not occupancy measures. While it is true that a pure strategy (a singleton stochastic policy) and mixed strategy (a distribution $\rho$ over many policies) can both induce the same occupancy measure, these strategies are not the same from the policy view (by definition). We discuss the importance of this discrepancy above Thm 1. In addition, we understand that optimal policies can always be deterministic (stated in line 78, col 2), however this does not mean that there always exists an NE that contains only deterministic policies (is that your concern?). Recall that cMGs generalize MGs which generalize normal-form games like rock-paper-scissors which famously only have an NE in stochastic policies.
>
> Baselines: Sim and RR represent, in fact, two ways of extending (variational) policy gradient (VPG) to the multi-agent setting. Zhang et al. ‘20 proved that VPG solves cMDPs. They study the model-free RL setting whereas we study the model-based setting where we can compute exact policy gradients per-player with differentiation.
>
> Complexity: The Bellman equation no longer holds in cMDPs and hence not in cMGs. Previous NE-existence results on MGs [Fink, ‘64] leveraged state value functions, which we cannot do here. We should emphasize the loss of Bellman optimality, so thank you for raising this issue. In addition, computing NEs in MGs is PPAD-hard [pg 3, “The Complexity of Markov Equilibrium in Stochastic Games”]; cMGs are at least as hard. Therefore, PGL expectedly lacks guarantees although it mimics a standard protocol for solving games (McKelvey & Palfrey, ‘95; ‘98; Gemp et al., ’22; Eibelshauser & Poensgen, ‘23). We expect we can give guarantees under certain assumptions (potential, zero-sum), but we leave that to future work. Note that many of the domains we study empirically (e.g., IPD) satisfy neither assumption.
>
> CURL Literature: Thanks for your pointers to cMDP/CURL research. This list also acutely highlights that, despite the strong interest in CURL, *no one* has studied the setting where $n$ CURL agents interact. We introduce the necessary scaffolding upon which to extend single agent CURL to their multi-agent analogues, similar to how Littman’s MGs generalized RL. Each of the references you provide constitutes an important single-agent advancement that could be interesting to examine in conjunction with other learning agents and we are happy to raise them as important directions for future work.
>
>
>
> Hazan vs Alternatives: First, note we credit Hazan in the intro. We generalize from cMDPs because they are convex programs (CPs) with properties critical to our proofs (the solution set of a CP is convex [Thm 1]; convex losses allow suboptimality bounds in terms of gradient norms [Thm 2]). In contrast, finite-trial convex RL is NP-hard in the single trial setting and for submodular RL, "the resulting optimization problem is hard to approximate".
>
> Tangent-space Projections: Consider a 1-state, 2-action MDP with policy $[p_1, p_2]$; each action earns +1 reward. The expected reward is $p_1 + p_2$ and the gradient is $[1, 1]$. Note that any policy is optimal, yet the norm of the gradient is $\sqrt(2)$. However, if we project the gradient onto the tangent space of the simplex (i.e., subtract the mean of the entries), we find it is $[1, 1] - [1, 1] = [0, 0]$ which has zero norm. For more info, see Linear and Nonlinear Programming by Luenberger, 1984, Sec 12.4, p 364.
>
> Empirical Support for PGL: Figure 3 demonstrates PGL reaches low $\epsilon$ (equiv. well approximates NEs) in 3 games. We continue to report low $\epsilon$ in the other games. We also discuss how PGL not only finds NEs, but ones with interesting behavioral properties (e.g., Tables 2, 3, 4).
>
> Model-based vs Model-free RL: We point out “model-free” RL as future work, but will further qualify that PGL is “model-based” in the intro. Note that our title highlights the cMG framework, not PGL. We are happy to point out “model-free” as an important direction for future work within the cMG framework.
>
> Minor: Yes, $r_i$ can be interpreted as a reward vector.

---

> > ### Comment · Reviewer_Bm64 · 2025-04-03
> >
> > Dear authors,
> >
> > Many thanks for getting back to me with thorough replies. I am providing a few follow-up comments below to make sure some of the points I raised are clear enough. I do not think any of my concern is a "deal-breaker": The paper is interesting and, although many aspects could be further studied, I may be applying an unfair standard here. I will engage with other reviewers and reconsider my evaluation.
> >
> > COMMENTS
> >
> > 1) Pure/Mixed strategies: The authors might be right, but the rock/scissor/paper example makes me think my point wasn't clear. Of course I see the difference in playing deterministic policies or stochastic policies. Indeed, I believe the right definition of pure strategy is "each player is playing a deterministic policy" and a strategy is mixed when some players are playing stochastic policies. However, according to the definition in the paper, players can play "pure strategies" with stochastic policies, while in mixed strategies they are playing mixture of stochastic policies. To me both look like mixed strategies: I am wondering if the paper is actually only analyzing mixed strategy and mixed strategy in disguise. The value of the game does not depend on the policies given the occupancy, which means that for every mixed strategy I have a pure strategy (of stochastic policies) inducing the same occupancy, hence the same value. Perhaps there is a difference on what happens with unilateral deviations from a fixed strategy: Even if the value is the same, deviating from a "pure" strategy of stochastic policies is different than deviating from a "mixed" strategy of mixtures of stochastic policies. I do not see it, so I am asking the authors if that is the case.
> >
> > 2) Hazan vs alternatives: This point would be stronger if the paper was giving an algorithm for exact NE, which clearly cannot be obtained for other formulations that are NP-hard even in the single-agent setting. However, the paper is giving an approximate solution. Finite-trial and submodular may also admit "approximate" solutions (not approximating the global optimum, just a "good" solution)...
> >
> > 3) Guarantees of PGL: The guarantees I was looking for are not empirical, but theoretical. Do we have any indication that PGL is guaranteed to converge to an equilibrium beyond empirical evidence in some domains?
> >
> > 4) Model-based vs model-free RL: I am worried I have to disagree here. RL is a framework for decision-making **from interactions**. Even in model-based RL, the model of the environment is learned from samples. If the model is known and the algorithm is not sample-based in any way, I do not think it shall be called RL.

---

> > > ### Author Response · Authors · 2025-04-03
> > >
> > > Thank you again for your comments and for your continued engagement.
> > >
> > > 1. Pure/Mixed strategies.
> > > We describe a cMG below in an effort to elucidate some of your concerns. We are assuming you have raised this discussion of definitions as it relates to the NE existence results, but please correct us if that assumption is incorrect. TL;DR: different policies are not generally exchangeable in games; even though different policies can induce the same occupancy measure (and same value), they induce different environments for the other players and so affect the NEs. Consider a cMG with two players. Player 1 stands on a safe podium and controls a switch. If the switch is on, the ground is lava. If the switch is off, the ground is safe. Player 2 stands on a safe spot and can observe the switch. Player 1’s utility is to maximize the long-run entropy of lava occurrence summed over the state-space as well as the entropy of player 2’s state distribution, i.e., aim for [50% lava, 50% no lava] and for player 2 to explore beyond their safe spot. Player 2 receives a small reward for staying in their safe spot, a very large negative reward for stepping on lava, a very large reward for collecting a treasure from outside their safe spot, and zero otherwise. Let player 2’s current policy be to never move from their safe spot. Consider two different policies (both with the same value) for player 1. In policy A (a mixture of deterministic policies), player 1 either turns the switch on at the beginning of the cMG or leaves it off. In policy B (a singleton stochastic policy), player 1 flips the switch with 50% probability at every time step. Both of these induce the same occupancy measure and also the same value for all players given player 2’s current policy. However, player 2’s best response to policy A, call it policy A’, might be to stay inside when they see the switch is on (because given player 1 plays A, they know it is for the whole episode), but when they see it is off (for the day), they venture out to collect the treasure. In contrast, player 2’s best response to policy B, call it policy B’, is likely to always stay inside and collect their mild reward. The takeaway is that although policy A and B induce the same occupancy measure, they actually have very different effects on player 2’s decision making. Both (A, A’) and (B, B’) are NE, however, (B, A’) and (A, B’) are not. It’s our understanding that you are concerned (B, A’) and (A, B’) might trivially exist as NE based on your reasoning, however, note that A and B are not exchangeable even though they achieve the same value. Prop 1 proves (somewhat trivially) that NEs of type (A, A’) exist (although not necessarily with such low support, e.g., mixtures of only 2 policies vs infinite policies). Theorem 1 proves NEs of type (B, B’) exist and requires more careful attention.
> > > 2. Note that although convex MDPs admit efficient approximate algorithms, cMGs, in general do not. Even computing $\epsilon$-NE for constant $\epsilon$ is PPAD-complete for n-player, general-sum: “Inapproximability of Nash Equilibrium”, Rubinstein ‘15.
> > > 3. No, PGL has no theoretical guarantees, but that is expected. Note that it is known that no dynamics converge to NE in n-player, general-sum: "An impossibility theorem in game dynamics", Milionis et al. '23.
> > > 4. We agree. PGL does not meet the precise definitions of a model-based RL method. We just mean it corresponds to the infinite sample limit of a model-based approach. We do not claim anywhere in the current draft that PGL is an RL method, so we will leave it that way.
> > >
> > > Thank you again for engaging with us and for your interesting questions. Since, as you state, you find the paper interesting, many aspects of which could be further studied, and none of your concerns is a deal breaker, could you please consider increasing your score accordingly.

---

### Official Review · Reviewer_yTwy · 2025-03-12

**Overall Recommendation:** 2

**Summary:**

The paper studies a generalized model of Markov games, called convex Markov games (CMG). The difference between CMGs and standard Markov games is that the former adopt convex functions as the players' utility functions, which are more general than linear functions used by the latter. More specifically, each player's utility is a convex function of the state-action occupancy probabilities. The paper studies n-player CMGs. It proved that a Nash equilibrium---in particular a pure one---always exists in every CMG. It then presented a projected-gradient loss minimization algorithm to compute an approximate equilibrium. Finally, experiements were conducted to evaluate the results, showing novel solutions that approximate human play.

**Claims And Evidence:**

The paper provided proofs and experiments to support the claimed results. The proofs look sound but could possibly be optimized a bit more.

**Essential References Not Discussed:**

No.

**Experimental Designs Or Analyses:**

The experiment designs look reasonable to me. I don't find any issues with this part.

**Methods And Evaluation Criteria:**

The methods used look reasonable to me.

**Other Comments Or Suggestions:**

I don't have any other comments besides what I commented above.

**Other Strengths And Weaknesses:**

Strength: Convex Markov games are a well-motivated model. The paper made valuable contributions towards understanding such games.

Weaknesses:

- The main theorem, Theorem 1, is an important result, but the proof relies heavily on the result by Kosowsky, which was intended for an even more general class of games. While I have no criticism about this proof approach, I feel it would still be nice to see if a simpler proof can be constructed without relying on the result by Kosowsky. A proof specifically for CMGs could possibly help us to understand CMGs better.

- A more critical issue is about the clarity of Section 4. The current presentation makes it hard to understand the computational approach. For example, while the authors introduced exploitability, what does it mean for the computing of a Nash? I think a high-level explanation about the general approach in several sentences could be helpful.

Additionally, what does the bound presented in Theorem 2 mean? Somewhat abruptly, the task reduces to minimizing the loss term in Equation (11). But even if this term is minimized, there is still a log term in the upper bound of $\epsilon_i(\pi)$. So what exactly is the relation between a minimum solution to this problem and an equilibrium? Does that mean the log term is unimportant?

In Theorem 2, $\pi$ is said to be an approximate equilibrium but without defining what an approximate equilibrium mean.

The part below Equation (12) is also hard to follow, without sufficient useful information about the intuition. The Opt operator in Line 3 of Algorithm 1 also requires more explanation. What specific optimization problem does it solve, with respect to $\pi$ and the gradient parameter?

In summary, while the overall structure and organization of the paper is reasonable and the problems studied are interesting and well-motivated, the paper could have been made more accessible and informative. There are some clarity issues and some parts lack rigorous definitions and necessary details.

**Questions For Authors:**

See Other Strengths And Weaknesses. There is no need to answer every question though; some of them are just examples of the weaknesses.

**Relation To Broader Scientific Literature:**

Convex Markov games are a natrual extension to Markov games, so they are closedly related to the areas of game theory, sequential decision making, and reinforcement learning.

**Theoretical Claims:**

The proofs look sound but could possibly be optimized a bit more.

---

> ### Author Rebuttal · Authors · 2025-04-01
>
> Dear reviewer, thank you for your constructive feedback. We are pleased to hear you find the proposed convex Markov Game model **“well-motivated”** and the proof of existence of pure Nash equilibria an **“important result”**. We appreciate the need for clearly explaining this important result and our proposed algorithm.
>
> Regarding NE existence, we will include a simplified version of Theorem 1’s proof based on Debreu that avoids some of the *topological* discussion of Kosowsky (although contractibility is still a key component). Note that unlike the NE existence proof for (vanilla) Markov games which makes use of the Bellman optimality of state-value functions and appears tailored “specifically for MGs”, Bellman’s equation no longer holds in convex MDPs and so this proof technique is not available to us. We understand your point though how a proof can help understand the domain, but, in this case, developing a more “RL-flavored” proof appears to us to be non-trivial.
>
> Thank you for pointing out these points of confusion in Section 4. Your comments will help us to improve the writing. We will do our best to answer your questions anyways in case you are able to let us know if they make the approach clearer.
>
> Approximate NE: NE is precisely a profile $x$ such that exploitability (9) equals zero. Every policy profile is technically an "approximate equilibrium"; $\epsilon$ measures the level of approximation. This is analogous to saying every policy is an "approximate" solution to an MDP, however, the level of approximation is what is relevant. If $\epsilon = 0$, the profile is no longer approximate, it is exactly an NE.
>
> High Level Approach: It should also be clear from the $\max$ in (10) that exploitability (9) is always non-negative. Therefore, one can imagine solving for an NE (where $\epsilon = 0$) by minimizing exploitability, i.e., using it as a loss function. Exploitability is non-convex (this is not obvious and we will add a comment) hence vanilla gradient descent is not guaranteed to find a global minimum. However, we leverage "temperature annealing" ideas that have been successful in several other game classes (NFG/EFG/MGs) and show they can be successful empirically for cMGs as well.
>
> Log Term: The log term is still important and appears again in (12), which relates the minimum of $\mathcal{L}^{\tau}$ (11) to an equilibrium (9). In order to shrink the log term we must decrease the temperature $\tau$, which is why we anneal $\tau$ with a schedule in experiments (see input to Algorithm 1).
>
> Opt Notation: The “Opt” notation in Algorithm 1 is meant to serve as a first-order optimization oracle, e.g., gradient descent, for minimizing $\mathcal{L}^{\tau_t}$. Opt($\pi, \nabla_{\pi} \mathcal{L}^{\tau_t}$) just means that the user supplies an initial policy $\pi$ and the gradient operator, and the first-order oracle performs several descent iterations before returning a new policy.

---

### Official Review · Reviewer_8Rey · 2025-03-19

**Overall Recommendation:** 4

**Summary:**

The authors introduce a class of convex Markov games that allow general convex preferences, prove that pure strategy Nash equilibria exists, and provide a gradient-based approach to approximate this equilibria, noting the computational difficulty of finding the actual equilibria with general solvers. The significance of this work is from extending convex MDPs to multi-agent settings and providing rigorous analysis to show the existence of NE.

**Claims And Evidence:**

Every proposition, lemma and theorem are accompanied by clear proofs and relevant references. It is evident that the authors dedicated significant time to examining various mathematical theorems (such as fixed-point theorems, convex programming properties, etc.) to substantiate their claims. Derivations do tend to skip quite a few steps, thought I understand that authors had to be succinct due to page limits. Since this manuscript is dense with notations and mathematical derivations, added more detail in the appendix would've been great.

**Essential References Not Discussed:**

N/A

**Experimental Designs Or Analyses:**

See comments under "Methods And Evaluation Criteria"

**Methods And Evaluation Criteria:**

The authors provide a detailed overview of their simulation domains, explain each aspect of the policy, and offer clear insights into how each policy characteristics should be interpreted. They provide brief descriptions of what "creativity", "fairness", etc., means, and their remarks under each subsection are thorough.

One comment as a reader is that, it took a few iterations to understand how each application is relevant to different aspects, and what the authors mean by "creative" or "fair" policies. For "creativity", I'm still not sure why the presented policy in the path finding domain is considered creative. For "fairness", I'm not sure why fair visitation of states is important in practice, since that may not always be a desired characteristic. I do think the "safety" subsection is convincing and easy to follow.

**Other Comments Or Suggestions:**

1. At the beginning of section 5, authors mentioned "four" baselines, and I see that they immediately list them out (min \epsilong, Sim, RR, and SGAME). But it looks like these four baselines are only mentioned under "creativity", so I just want to note that I was a little confused when going through the rest of section 5. I personally would've been okay with the mention of baselines for each specific subsection.
2. Page 6, in the last sentence of the first paragraph under "Remark", should it be just "player" or "players" instead of "player's"?

**Other Strengths And Weaknesses:**

The authors did a great job pointing out the gap in literature, and highlighting their contribution from early on. By doing so, the authors set the tone for the manuscript, and made it easier for the reader understand the manuscript's purpose and relevance. The paper is well-structured, with the authors effectively establishing the theoretical foundation through clear proofs and intuitive explanations of how each expression is derived. They then offer a thorough and insightful analysis of their simulations. See comments under "Claims And Evidence" and "Methods And Evaluation Criteria" for additional detail.

Meanwhile, I do fee that too much information was front-loaded in section 5. It may be due to how I process information, but I felt that not all information provided at the beginning were necessary for each subsection, so deferring some for specific subsections would've been better for readability.

**Questions For Authors:**

None.

**Relation To Broader Scientific Literature:**

Extending convex MDPs to multi-agent settings enables scalable approaches to handling the complex interactions between multiple agents. This extension provides a foundation for designing more robust and tractable solutions in settings with multiple decision-makers.

**Theoretical Claims:**

I reviewed all derivations in the main body, and proofs of theorems 1, 2, and 3 in the appendix. See comments under "Claims And Evidence".

---

> ### Author Rebuttal · Authors · 2025-04-01
>
> Dear reviewer, thank you for your constructive feedback. We are glad to hear that the significance of our work carried through and that you think we **“did a great job pointing out the gap in literature”** (convex MDP + multi-agent). We also understand your comment regarding the layout of the experiments section and can see how “front loading” so much context can disrupt the flow and actually make the remaining subsections focused on each domain harder to understand. We will move (or copy) some of this information to each subsection where it is immediately relevant to the reader. We will also add more detail to the appendix, in particular, adding an alternative (arguably simplified) Nash equilibrium existence proof based on Debreu ‘52 rather than Kosowsky ‘23.
>
> Regarding “creativity” and “fairness” terminology: The terminology "creative" is borrowed from prior work (Zahavy et al., ‘23). The rationale is that adding the entropy bonus encourages exploring the whole state-action space so the agents can possibly uncover new "creative" solutions. We can add this explanation to clarify. The “fairness” experiment looks at fair visitation of the plays “Bach” and “Stravinsky”; this game, also known as “Battle of the Sexes” in the classical literature, exhibits two pure equilibria where only one player’s preferred outcome is achieved and one mixed equilibrium where both outcomes are achieved in expectation. We hope our fairness metric makes sense in that context, but we agree “fairness” of state-visitation is not suitable in every domain.
>
> Minor: Good catch. “player’s” should be “players”.

---

### Decision · Program_Chairs · 2025-05-01

**Decision:**

Accept (poster)

**Comment:**

The paper studies a new framework of convex Markov games, which generalizes the players' linear utilities with any convex function of their state-action occupancy measures, following the recent similar extension in the single-agent setting for MDPs. Both the existence and the computation results were presented for finding an approximate equilibrium of the game. Some experiments were also presented to validate the effectiveness of the algorithms. There was some concern regarding the novelty of the techniques, which is valid. Overall, the model is new, and the results are of merit to the multi-agent RL domain. I recommend the authors to incorporate the feedback from the reviewers in preparing the camera ready version of the paper.